# Residual Kernel Policy Network: Enhancing Stability and Robustness in RKHS-Based Reinforcement Learning

**Yixian Zhang, Huaze Tang** *& **Huijing Lin**
Tsinghua Shenzhen International Graduate School, Tsinghua University
{yixian-z24,tanghz24,linhj24}@mails.tsinghua.edu.cn

**Wenbo Ding**[†]
Tsinghua Shenzhen International Graduate School, Tsinghua University
Shanghai Artificial Intelligence Laboratory
ding.wenbo@sz.tsinghua.edu.cn

## Abstract

Achieving optimal performance in reinforcement learning requires robust policies supported by training processes that ensure both sample efficiency and stability. Modeling the policy in reproducing kernel Hilbert space (RKHS) enables efficient exploration of local optimal solutions. However, the stability of existing RKHS-based methods is hindered by significant variance in gradients, while the robustness of the learned policies is often compromised due to the sensitivity of hyperparameters. In this work, we conduct a comprehensive analysis of the significant instability in RKHS policies and reveal that the variance of the policy gradient increases substantially when a wide-bandwidth kernel is employed. To address these challenges, we propose a novel RKHS policy learning method integrated with representation learning to dynamically process observations in complex environments, enhancing the robustness of RKHS policies. Furthermore, inspired by the advantage functions, we introduce a residual layer that further stabilizes the training process by significantly reducing gradient variance in RKHS. Our novel algorithm, the Residual Kernel Policy Network (ResKPN), demonstrates state-of-the-art performance, achieving a 30% improvement in episodic rewards across complex environments.

## 1 Introduction

Reproducing Kernel Hilbert Space (RKHS) methods have emerged as powerful tools in reinforcement learning (RL) due to their ability to model policies nonparametrically, allowing for flexible function approximation and efficient exploration of the solution space (Lever & Stafford, 2015; Paternain et al., 2020). By leveraging kernels, RKHS-based policies can capture complex relationships in high-dimensional observation spaces, leading to expressive models that adapt well to diverse environments (Paternain et al., 2022). However, despite these advantages, RKHS-based policies face significant challenges that limit their practical applicability. A critical issue is the high variance in policy gradients inherent to RKHS methods (Smith & Egeland, 2024). This excessive variance arises because the RKHS gradient updates do not fully exploit all previously sampled episodes, leading to instability during training and difficulties in converging to optimal policies (Dastider et al., 2022). Additionally, the robustness of RKHS models is often compromised due to hyperparameter sensitivity. Different environments may require distinct hyperparameter settings, making it challenging for RKHS policies to maintain consistent performance across varied and complex environments (Liu & Lian, 2024). In environments with large action spaces, such as the Humanoid environment (Todorov et al., 2012), RKHS methods struggle to scale effectively.

---

*Contribute equally to this work.
[†]Corresponding author

From our review of existing approaches, grid search methods are commonly employed for hyperparameter tuning to align the data distribution with the RKHS kernel (Montesinos López et al., 2022; Hsu & Lin, 2002; Wilson et al., 2016). The requirement for meticulous hyperparameter tuning introduces additional computational complexity and limits the adaptability of RKHS policies to new tasks. Regarding variance reduction, various techniques such as learning rate search (Le et al., 2019), symmetric estimation (Paternain et al., 2020), and policy search (Chen et al., 2016) are proposed to stabilize the training process, while the effectiveness of these variance reduction methods is often constrained to specific environments. Alternatively, variance can be mitigated by introducing bias into the estimation process, such as leveraging predefined kernel orthogonal basis (Mazoure et al., 2020) or employing an online clustering approach to aggregate similar kernel orthogonal bases into central representations (Wang & Principe, 2021). Despite their promise, the implementation of these methods in high-dimensional spaces poses significant challenges due to computational inefficiencies.

To fully leverage the advantages of RKHS methods and develop stable, robust policies across diverse environments, we introduce the **Residual Kernel Policy Network (ResKPN)**. Our approach integrates representation learning with RKHS policy models to dynamically process observations, enhancing the adaptability and robustness of RKHS policies in complex environments. By incorporating a neural network for feature extraction, we adjust the distribution of inputs to better align with the chosen kernel, mitigating hyperparameter sensitivity. Furthermore, inspired by the variance reduction capabilities of advantage functions, we introduce a residual layer (He et al., 2016b) that significantly reduces gradient variance within RKHS. This addition stabilizes the training process and enables the discovery of high-performing policies with improved robustness. Specifically, the following key contributions are made:

- We propose a novel RKHS policy learning algorithm that employs a neural network to dynamically represent observations, enhancing the adaptability and robustness of RKHS policies across diverse environments by aligning observation distributions with the chosen kernel.

- We conduct an in-depth analysis of the high variance issue in RKHS policy gradients. Our findings reveal that learning with traditional RKHS policies, particularly in wide-bandwidth kernels, leads to significant instability and high variance during training.

- We introduce a variance reduction technique by designing a residual layer for the RKHS policy. Our analysis demonstrates that this approach effectively reduces gradient variance and stabilizes the training process. Combined with representation learning, our ultimate algorithm, ResKPN, achieves superior performance across various challenging environments, including a 30% episodic rewards improvement in the Humanoid environment.

## 2 BACKGROUND

In this section, we review previous studies on reinforcement learning within RKHS. Additionally, we provide a brief overview of variance reduction techniques in reinforcement learning to facilitate the subsequent introduction of the variance reduction methods designed for RKHS policies. To describe the methodology of the ResKPN algorithm, we also examine prior research on the integration of kernel methods and deep learning techniques.

### 2.1 REINFORCEMENT LEARNING WITHIN RKHS

**Reinforcement learning** algorithms attempt to learn the optimal $Q$-function for the cumulative rewards or the optimal policy (Sutton & Barto, 2018). This paper concentrates on the latter one, which leads to the policy gradient algorithm. The cumulative rewards (Bedi et al., 2024) is define as

$$U(\pi_w) = \mathbb{E}_{\tau \sim p(\tau;\pi_w)} \left[ \sum_{t=1}^{\infty} \gamma^{t-1} r\left(s_t, a_t\right) \right],$$

where $p\left(\tau; \pi_w\right) = p(s_1) \prod_{t=1}^{\infty} p(s_{t+1}|s_t, a_t)\pi_w(a_t|s_t)$ represents the distribution of the trajectory $\tau = ((s_1, a_1), (s_2, a_2), ...)$ following the policy $\pi_w$ and $r(s_t, a_t)$ denotes the instant reward of state-action pair $(s_t, a_t)$. $\gamma$ is the discounted factor of reward. The policy gradient is to find the gradient

direction to maximize $U(\pi_w)$ that

$$\nabla_w U(\pi_w) = \frac{1}{1-\gamma} \mathbb{E}_{a \sim \pi(a|s), s \sim \rho_{\pi_w}(s)} \left[ Q^{\pi_w}(a,s) \nabla_w \log \pi_w(a|s) \right], \tag{1}$$

where $Q^{\pi_w}(a,s) = \mathbb{E}_{\tau \sim p(\tau; \pi_w)} \left[ \sum_{t=1}^{\infty} \gamma^{t-1} r(a_t, s_t) | a_1 = a, s_1 = s \right]$ is the $Q$-function and $\rho_{\pi_w}(s)$ represents the marginal density of the state under policy $\pi_w$.

**Reproducing Kernel Hilbert Space** (RKHS) is the vector valued Hilbert Space $\mathcal{H}_K$ where an elements $K(x, \cdot) \in \mathcal{H}$ satisfies the reproducing property $\langle K(x, \cdot), K(y, \cdot) \rangle = K(x, y)$. Despite the policy is modeled by the parameter $w$ with particular parameterized functions, the stochastic policy is directly modeled as a function $h$ in RKHS $\mathcal{H}_K$, where the updating gradient for it is also a function. In detail, the action is chosen from a multivariate normal distribution $\mathcal{N}(h(s), \Sigma)$

$$\pi_{h, \mathbf{\Sigma}}(a|s) := \frac{1}{Z} e^{-\frac{1}{2}(h(s)-a)^\top \mathbf{\Sigma}^{-1}(h(s)-a)}, \tag{2}$$

where the mean value is dependent on the function $h \in \mathcal{H}_K$. The gradient for the RKHS policy is then derived (Paternain et al., 2020; Lever & Stafford, 2015) as

$$\nabla_h U(\pi_h) = \frac{1}{1-\gamma} \mathbb{E}_{a,s} \left[ Q^{\pi_h}(a,s) K(s, \cdot) \mathbf{\Sigma}^{-1}(a - h(s)) \right],$$

where the derivative with respect to function $h$ uses the Fréchet derivative (Mcgillivray & Oldenburg, 1990). To compute the stochastic gradient $\nabla_h U(\pi_h)$, a common approach is to use Monte Carlo approximation (Lever & Stafford, 2015), which incurs high computational costs, particularly in complex environments. In (Pontil et al., 2005; Paternain et al., 2020), a pair $(s, a)$ is sampled from trajectories to obtain an unbiased estimate of $\nabla_h U(\pi_h)$. The accuracy of this estimate depends on the precision of the $Q$-function approximation $Q^{\pi_h}(z)$, for which actor-critic methods (Dastider & Lin, 2022) are employed to ensure more stable estimations. In this paper, the estimation of $\nabla_h U(\pi_h)$ is obtained from the one pair estimation in (Cervino et al., 2021)

$$\nabla_h \hat{U}(\pi_h) = \eta K(s_k, \cdot) \mathbf{\Sigma}^{-1}(a_k - h(s_k)) \hat{Q}^{\pi_h}(a_k, s_k), \tag{3}$$

where $\eta$ represents the learning rate.

## 2.2 VARIANCE REDUCTION AND DEEP KERNEL LEARNING

**Variance in policy gradient** is seen as the main factor influencing the performance of policy gradient algorithms (Hafner & Riedmiller, 2011). Aiming to reduce the variance, a series of classic reinforcement learning algorithms are proposed. Designed with the famous actor-critic framework (Grondman et al., 2012), baseline control variate is proposed for leveraging the vibration in $Q$-function (Greensmith et al., 2004). It introduces a baseline function, which is mainly chosen as the state value function $V(s) = \mathbb{E}_a[Q(a,s)]$, to eliminate unnecessary variance introduced by state values, which formulates as

$$\nabla_w \hat{U}(\pi_w) = A^{\pi_w}(a_k, s_k) \nabla_w \log \pi_w(a_k|s_k),$$

where $A^{\pi_w}(a_k, s_k) = Q^{\pi_w}(a_k, s_k) - V^{\pi_w}(s_k)$ is known as the advantage function. The update size of the policy gradient is further limited in the generalized advantage estimation method (Schulman et al., 2016) by introducing an iteration optimization algorithm for trust region policy optimization (Schulman et al., 2015). Combining the well-performance variance techniques, the proximal policy optimization (PPO) algorithm is proposed (Wu et al., 2021), achieving the overall best performance, which is viewed as the baseline algorithm in the successive research. The ultimate gradient $\nabla_w U(\pi_w)$ in PPO is formulated as

$$A^{\pi_w}(a_k, s_k) \min \left( \frac{\pi_w(a_k|s_k)}{\pi_{w_-}(a_k|s_k)}, \text{clip} \left( \frac{\pi_w(a_k|s_k)}{\pi_{w_-}(a_k|s_k)}, 1 - \epsilon, 1 + \epsilon \right) \right) \nabla_w \log \pi_w(a_k|s_k), \tag{4}$$

where $\pi_{w_-}(a|s)$ represents the policy in the last iteration, and $\epsilon$ is the ratio of the clip, limiting the update size of the new policy. Based on this, the discovered policy optimisation (DPO) algorithm (Lu et al., 2022) is designed to further smooth the training process while encouraging the exploration, ahieving the overall best performance.

**Deep Kernel Learning** is widely studied with the development of kernel methods and neural networks. Kernel methods can learn a wide range of conditional distributions and predictive functions conditioned on context sets of arbitrary sizes (Kim et al., 2019). However, their applicability is often constrained by the necessity of designing task-specific kernels. To address this limitation, deep learning techniques are introduced to parameterize observations through learnable network layers, giving rise to deep kernel learning (Wilson et al., 2016), which enhances the scalability and performance of the model on complex tasks. For example, (Papamarkou et al., 2024) employs deep kernel processes to improve digit classification accuracy, while (Kristiadi et al., 2020) incorporates kernels into ReLU networks for more efficient predictions. Moreover, wide residual networks can be interpreted as performing kernel regression within the associated RKHS (Lai et al., 2023), offering a theoretical explanation for the smoother functions they learn. This smoothness contributes to the networks' superior generalization capabilities (Tirer et al., 2022).

Despite the strong performance of deep kernel learning in typical deep learning tasks, few studies integrate deep learning methods with kernels in RKHS for policy gradient. To the best of our knowledge, this is the first paper to incorporate representation learning through neural networks into RKHS policy gradient methods for the rapid identification of locally optimal policies. We present a novel formulation of the RKHS policy gradient and provide a detailed analysis of the algorithm's stability.

## 3 PROPOSED METHOD: THE RESIDUAL KERNEL POLICY NETWORK

In this section, we introduce the proposed RKHS policy gradient method, ensuring a stable training process across multiple environments. We begin by analyzing the limitations in the current RKHS policy gradient: the insufficient representational capacity and the excessive variance. In order to enhance representational capacity for learning in complex environments, we describe representation learning in Section 3.2. This representation learning is integrated with a neural network updated using the Proximal Policy Optimization (PPO) algorithm, enabling the adaptive adjustment of observation distributions across different environments. In Section 3.3, we introduce advantage functions to reduce the variance in the RKHS gradient. Additionally, based on the representation learning, we design a residual layer as the baseline function for the RKHS policy to further minimize variance.

### 3.1 THE LIMITATIONS IN RKHS POLICY GRADIENT

In this section, a detailed analysis is conducted to illustrate two main drawbacks in the RKHS policy gradient: the insufficient representational capacity (Wilson et al., 2016) and excessive variance (Le et al., 2019).

**Insufficient representational capacity** is the primary issue limiting the learning adaptability of kernel methods (Wu & Wang, 2009). The hyperparameters of a kernel are highly sensitive to the distribution of input data, leading to under-fitting in uneven data distributions (Wang et al., 2020). A straightforward method to select hyperparameters is grid search (Hsu & Lin, 2002), while the computational complexity limits its use in high dimensional spaces. To adaptively learn the hyperparameters, they are modeled with linear functions or separation index. In (Wilson et al., 2016), the input of kernel is directly learned by neural networks, which achieves excellent performance in orientation extraction and magnitude recovery tasks. To illustrate the insufficient representational capacity of RKHS policies, we investigate their learning performance under different hyperparameter settings. Specifically, we employ the Gaussian kernel defined as $K(x,y) = \exp\left(-\frac{\|x-y\|^2}{\sigma^2}\right)$ to learn from observations, applying the gradient as described in Equation (3). As demonstrated in Figure 1a, the RKHS policy fails to update when $\sigma^2 = 0.01$ and $5.0$. In contrast, episodic returns improve during training for $\sigma^2 = 0.05, 0.1, 0.5$, and $1.0$. To the best of our knowledge, there is currently no general method for performing RKHS policy learning across various environments that offers strong robustness and insensitivity to hyperparameters.

**Excessive variance** is a prevalent issue in policy gradient methods (Greensmith et al., 2004). When applying the RKHS policy gradient, the problem becomes more prominent. We now compare the variance of gradient between the RKHS policy and the linear policy $\pi_\theta(a|s)$ defined in Appendix.

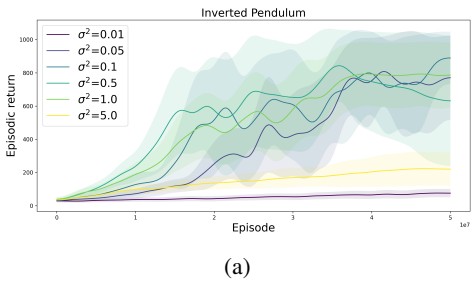 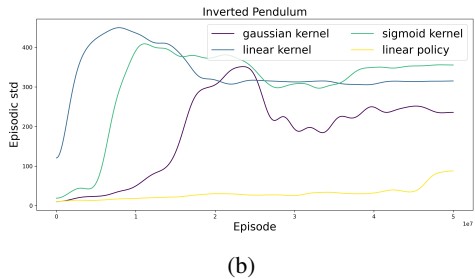

Figure 1: The insufficient representational capacity and excessive variance in RKHS policy gradient within Inverted Pendulum environment. (a) The learning performance choosing different $\sigma^2$. (b) The variance in training for different kernels compared with linear policy.

A.1, where the corresponding estimated gradient is denoted as $\nabla_\theta \hat{U}(\pi_\theta)$. We can prove that the variance ratio $\mathcal{R} = \frac{\mathrm{Var}_{a_k}(\nabla_h \hat{U}(\pi_h))}{\mathrm{Var}_{a_k}(\nabla_\theta \hat{U}(\pi_\theta))}$ is highly dependent on the choice of kernel $K(\cdot, \cdot)$.

**Lemma 3.1** *(A.1) Assuming that $\hat{Q}^{\pi_\theta}(a, s)$ and $\hat{Q}^{\pi_h}(a, s)$ conditioned on $s_k$ are linear functions, i.e., $\hat{Q}^{\pi_\theta}(a, s_k) = c_1^\top a + d_1, \hat{Q}^{\pi_h}(a, s_k) = c_2^\top a + d_2$, where $c_1, d_1, c_2, d_2$ are learnable parameters. Then, when the dimension of the action space is 1, the variance ratio $\mathcal{R}$ satisfies:*

$$\mathbb{E}_{s_k}\left[\frac{\mathrm{Var}_{a_k}(\nabla_h \hat{U}(\pi_h))}{\mathrm{Var}_{a_k}(\nabla_\theta \hat{U}(\pi_\theta))}\right] \geq \mathbb{E}_{s_k}\left[\frac{K^2(s_k, s_k)}{s_k^2}\frac{2c_1^2}{2c_2^2 + \Sigma^{-1}(c_2\theta^\top s_k + d_2)^2}\right].$$

We prove Lemma 3.1 in Appendix. A.1. Following this, it is observed that the variance of the RKHS policy is highly dependent on the chosen kernel, which leads to excessive variance when a wide-bandwidth kernel is selected. We test three different kernels in the Inverted Pendulum environment, comparing them with linear policy in Figure 1b. The results illustrate that the variance of all three kernels is significantly higher than that of linear policy, which coincides with the findings in Lemma 3.1.

## 3.2 LEARNING REPRESENTATIONS FOR RKHS KERNEL

The neural networks enable the learning model to extract multiple features from data using the back-propagation algorithm (LeCun et al., 2015). In order to adjust the data distribution for compatibility with the RKHS kernel, observations are initially input into a neural network for feature representation, and the subsequently distribution-adjusted representations are utilized for gradient iteration in the RKHS policy. Without loss of generality, we use $\psi_\vartheta(\cdot)$ to represent the neural networks with parameters $\vartheta$ for representation learning, where the RKHS policy in Equation (2) is adapted as

$$\pi_{\vartheta, h, \Sigma}(a|s) := \frac{1}{Z}e^{-\frac{1}{2}(h(\psi_\vartheta(s))-a)^\top \Sigma^{-1}(h(\psi_\vartheta(s))-a)}. \tag{5}$$

For the sake of clarity, the parameters of the policy are defined as $\varpi = (\vartheta, h, \Sigma)$. We update $\psi_\vartheta(\cdot)$ using the actor-critic scheme similar to the PPO algorithm, the critic $V_\delta^{\pi_\varpi}$ is also modeled as neural networks with parameters $\delta$, where the gradient derived from $TD$ error is formulated as

$$\nabla_\delta \widehat{TD}(\pi_\varpi) = \frac{1}{N}\sum_{t=1}^N \nabla_\delta\left[\left(\hat{V}_\delta^{\pi_\varpi}(s_t) - \hat{R}_t\right)^2\right],$$

where $\hat{R}_t$ is the target value (Wu et al., 2021). We use the operator $\mathcal{T}_{\pi_\varpi}(a_k, s_k)$ to represent the minimum calculation $\min\left(\frac{\pi_\varpi(a_k|s_k)}{\pi_{\varpi_-}(a_k|s_k)}, \mathrm{clip}\left(\frac{\pi_\varpi(a_k|s_k)}{\pi_{\varpi_-}(a_k|s_k)}, 1-\epsilon, 1+\epsilon\right)\right)$ in Equation (4). Based on the chain rule in derivative, the gradient of the neural networks $\psi_\vartheta(\cdot)$ is also derived as

$$\nabla_\vartheta \hat{U}(\pi_\varpi) = A^{\pi_\varpi}(a_k, s_k)\mathcal{T}_{\pi_\varpi}(a_k, s_k)\Sigma^{-1}(a_k - h(\psi_\vartheta(s_k)))\nabla_\vartheta h(\psi_\vartheta(s_k)).$$

This formulation resembles the loss function of the PPO algorithm, differing primarily in the inclusion of the RKHS function $h(\psi_\vartheta(s_k))$, where $h$ is updated using the RKHS gradient with the only modification being that the state $s_k$ is represented by the neural network $\psi_\vartheta(s_k)$:

$$\nabla_h \hat{U}(\pi_\varpi) = \eta K(\psi_\vartheta(s_k), \cdot) \mathbf{\Sigma}^{-1}(a_k - h(\psi_\vartheta(s_k))) \hat{Q}^{\pi_\varpi}(a_k, s_k). \tag{6}$$

Following the gradient estimation, the Kernel Policy Network (KPN) algorithm is shown in Algorithm 1. It should be noticed that the $h$ function only updates for $L$ times to avoid the explosion of the gradient. Meanwhile, the update epoch is set to update parameters of neural networks for $J$ times in each training, ensuring sufficient learning of critic and representation networks. With the integration of representation neural networks $\psi_\vartheta(\cdot)$, the RKHS policy can update dynamically during each training epoch. Experiment results show that the KPN algorithm achieves further better performance than pure RKHS policy even with tuned hyperparameters. Nevertheless, the excessive variance problem still exists, leading to the extreme reward cliff (Sullivan et al., 2022) in the training process.

---

**Algorithm 1** KPN algorithm

---

**Hyperparameters:** Total number of training steps $L$, mini-batch size $N$, step times of agent each training $T$ and the update epoch $J$ of critic and neural networks $\psi_\vartheta(\cdot)$ each training.

1: Initialize critic $V_\delta^{\pi_h}(s)$, RKHS function $h = 0$, actor $\pi_\varpi$ and replay buffer $\mathcal{B}$.
2: Start with the initial state $s_0$
3: **for** $l = 1, \ldots, L$ **do**
4:      Using policy $\pi_\varpi$, collect and store transitions $(s_t, a_t, r_t, s_{t+1})$ in replay buffer $\mathcal{B}$.
5:      **for** $j = 1, \ldots, J$ **do**
6:          Sample mini-batch $\{(s_i, a_i, r_i, s_{i+1}) \mid i = 1, \ldots, N\}$ from $\mathcal{B}$.
7:          Estimate the critic gradient $\nabla_\delta \widehat{TD}(\pi_\varpi)$ and update the parameters $\delta$.
8:          Estimate the neural networks $\psi_\vartheta(\cdot)$ gradient $\nabla_\vartheta \hat{U}(\pi_\varpi)$ and update the parameters $\vartheta$.
9:      **end for**
10:      Sample a transition $(s_i, a_i, r_i, s_{i+1})$ from $\mathcal{B}$.
11:      Estimate the RKHS gradient $\nabla_h \hat{U}(\pi_\varpi)$ and update the function $h = h + \nabla_h \hat{U}(\pi_\varpi)$.
12: **end for**

---

### 3.3 THE RESIDUAL LAYER FOR VARIANCE REDUCTION

In this section, we introduce two proposed variance reduction methods designed for the KPN algorithm, attaining a stable training process.

**The advantage function** is widely used as a surrogate for the $Q$-function in policy gradient methods. It is proven in (Weaver & Tao, 2001) that using the advantage function does not introduce bias into the estimation of the policy gradient $\nabla_w U(\pi_w)$. In this paper, we introduce the advantage function to decrease the variance for RKHS policy gradient $\nabla_h \hat{U}(\pi_\varpi)$ in the KPN algorithm. We adopt the gradient estimation in Equation 6 as:

$$\nabla_h \hat{U}_A^{\pi_\varpi}(\pi_\varpi) = \eta K(\psi_\vartheta(s_k), \cdot) \mathbf{\Sigma}^{-1}(a_k - h(\psi_\vartheta(s_k))) A^{\pi_\varpi}(a_k, s_k), \tag{7}$$

where $A^{\pi_\varpi}(a_k, s_k) = Q^{\pi_\varpi}(a_k, s_k) - V^{\pi_\varpi}(s_k)$. We denote the introduced algorithm with advantage functions as AdvKPN. Moreover, based on the representation networks in KPN algorithm, the residual layer is specifically designed to diminish the variance in RKHS gradient.

**The residual layer** is initially introduced in resnet structure to address the vanishing gradient problem (He et al., 2016b). The key idea of the residual layer is to learn the additive residual function $\mathcal{F}(x) + f(x)$, where $f(x)$ is usually chosen as the identity mapping $f(x) = x$. It is assumed that when the $x$ is the optimal value, $\mathcal{F}$ will converge to the zero mapping to skip the network layer (He et al., 2016a). In this paper, inspired by the convergence property of the residual layer, we also design a fully connected layer for the AdvKPN algorithm to form the final ResKPN algorithm. The complete algorithmic scheme is illustrated in Figure 2, with step-by-step details provided in the subsequent text. The policy integrated with the residual layer is expressed as follows:

$$\pi_{\varpi,\iota}(a|s) := \frac{1}{Z} e^{-\frac{1}{2}(h(\psi_\vartheta(s)) + \mu_\iota(\psi_\vartheta(s)) - a)^\top \mathbf{\Sigma}^{-1}(h(\psi_\vartheta(s)) + \mu_\iota(\psi_\vartheta(s)) - a)},$$

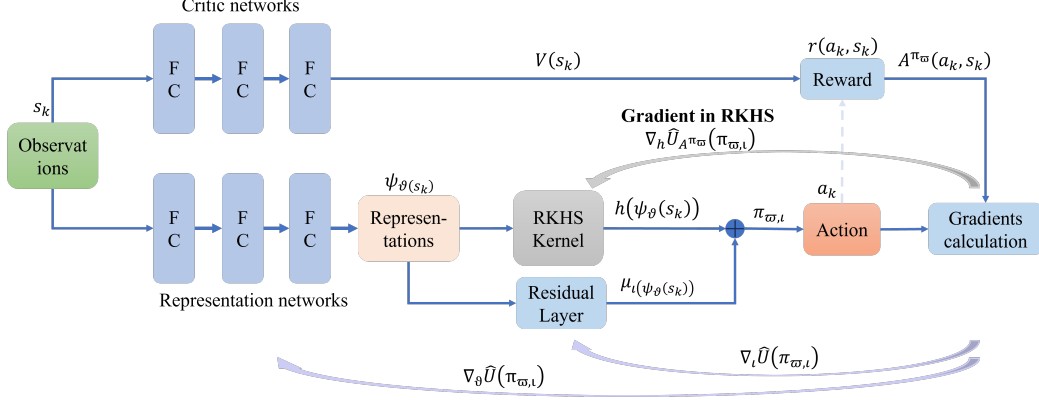

Figure 2: The scheme of the ResKPN algorithm.

where $\mu_\iota(\cdot)$ is the fully connected layer with parameters $\iota$. Therefore, it is easily derived that our ultimate RKHS gradient:

$$\nabla_h \hat{U}_{A^{\pi\varpi}}(\pi_{\varpi,\iota}) = \eta K(\psi_\vartheta(s_k), \cdot)\boldsymbol{\Sigma}^{-1}(a_k - h(\psi_\vartheta(s_k)) - \mu_\iota(\psi_\vartheta(s_k)))A^{\pi\varpi}(a_k, s_k). \quad (8)$$

It should be noticed that the gradient of the representation network is also changed to $\nabla_\vartheta \hat{U}(\pi_{\varpi,\iota})$ due to the integration of residual layer:

$$A^{\pi\varpi}(a_k, s_k)\mathcal{T}_{\pi_\varpi}(a_k, s_k)\boldsymbol{\Sigma}^{-1}(a_k - h(\psi_\vartheta(s_k)) - \mu_\iota(\psi_\vartheta(s_k)))\nabla_\vartheta(h(\psi_\vartheta(s_k)) + \mu_\iota(\psi_\vartheta(s_k))). \quad (9)$$

Following this, the ResKPN algorithm is obtained by substituting the gradient of RKHS and the representation network in Algorithm 1 with Equation (9) and Equation (8). The advantage value $A^{\pi\varpi}(a_k, s_k)$ is computed by combining the state value $V(s_k)$, represented by the critic networks, with the reward $r(a_k, s_k)$. The detailed methodology, along with supplementary explanations for Figure 2, is provided in Appendix B.1. Meanwhile, we can prove that the variance of the RKHS gradient above follows the following order:

**Theorem 3.2** *(A.2) Assuming that $Q^{\pi\varpi}(a_k, s_k) \geq \frac{1}{2}\mathbb{E}_{a_k}\left[Q^{\pi\varpi}(a_k, s_k)\right] = \frac{1}{2}V^{\pi\varpi}(s_k)$ and $a_k - h(\psi_\vartheta(s_k)) \geq \frac{1}{2}\mu_\iota(\psi_\vartheta(s_k))$ when they following the same policy $\pi_{\varpi,\iota}$, the variance of the RKHS policy gradient gradually decreases as:*

$$\mathrm{Var}_{a_k}(\nabla_h \hat{U}(\pi_\varpi)) \geq \mathrm{Var}_{a_k}(\nabla_h \hat{U}_{A^{\pi\varpi}}(\pi_\varpi)) \geq \mathrm{Var}_{a_k}(\nabla_h \hat{U}_{A^{\pi\varpi}}(\pi_{\varpi,\iota}))$$

We prove Theorem 3.2 in Appendix. A.2. The introduction of the residual layer is motivated by the variance reduction properties observed in the advantage function. Specifically, by incorporating a baseline function $V^{\pi\varpi}(s_k) = \mathbb{E}_{a_k}[Q^{\pi\varpi}(a_k, s_k)]$, which exhibits greater stability compared to $Q^{\pi\varpi}(a_k, s_k)$, the variance during training can be decreased. Similarly, for RKHS policies, a stable baseline function can be leveraged to reduce the variance of the RKHS gradient, which is shown in the proof of Theorem 3.2. An ideal candidate for this baseline is the residual network $\mu_\iota(\cdot)$. Furthermore, the integration of a fully connected layer enhances representation learning, and together, these two components jointly contribute to ResKPN's superior overall performance. To directly illustrate the variance reduction effect of this baseline function, an intuitive comparison is provided in Appendix C.

Based on the aforementioned adaptation, the RKHS policy retains non-parametric representations suitable for high-dimensional environments and enhances its representational capacity. Meanwhile, the excessive variance in the RKHS policy gradient is mitigated, resulting in a more stable learning process. The incorporation of a residual layer introduces an additional learner, leveraging the advantages of ensemble learning as well. In order to achieve the overall best performance, the techniques in PPO and DPO algorithms are also integrated into all algorithms proposed above to stabilize the training process, where the technique details are described in Appendix B.1.

## 4    EXPERIMENTAL RESULTS

In this section, various experiments are employed to verify the effectiveness of the representation learning and the variance reduction residual layer. We divide this section into three parts: (1) the environment setting of our experiment, (2) the comparison results including ablation study across multiple environments for the aforementioned algorithms, and (3) the analysis of the effectiveness of the proposed methods.

### 4.1    EXPERIMENT SETTING

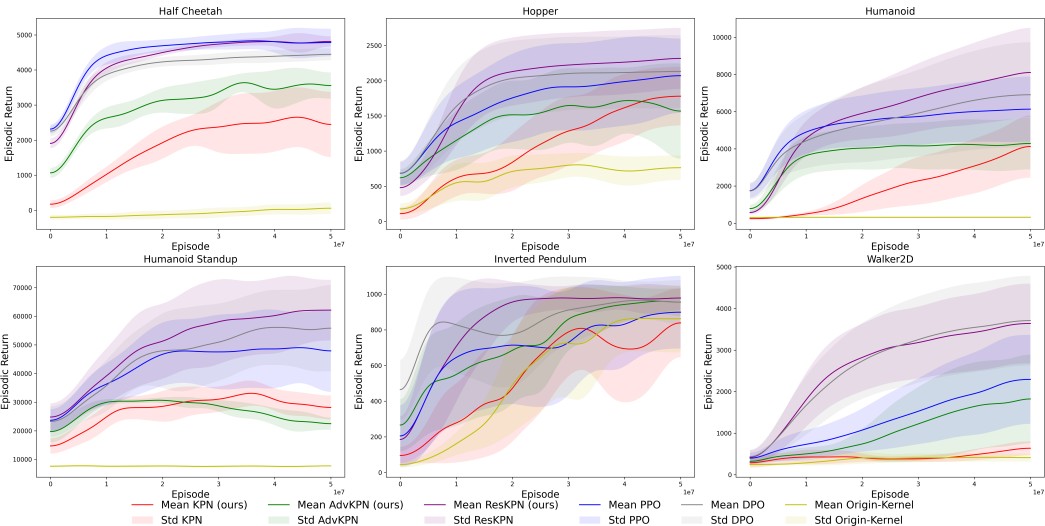

Figure 3: The episodic reward in multiple MuJoCo environments for proposed algorithms.

We evaluate our proposed algorithm on six continuous control tasks from the MuJoCo environments (Todorov et al., 2012), including the Inverted Pendulum, Hopper, Half Cheetah, Walker2D, Humanoid, and Humanoid Standup. These tasks are ordered by increasing complexity, ranging from low to high. The primary objective of these environments is to optimize the performance of simulated robotic agents to enhance the forward progress and maintain stability with minimized energy consumption (Batra et al., 2024). The detailed setup of our experiment is described in Appendix B.

### 4.2    COMPARISONS

We compare our results with the original RKHS policy (Origin-Kernel), DPO algorithm and the widely used baseline PPO algorithm in the comparison. The episodic reward of the training agents serves as the primary metric for assessing the effectiveness of the algorithms. We show the training performance of different methods in Figure 3.

As illustrated in the figure, our ultimate model, ResKPN, achieves the overall best performance. It converges faster than other baseline algorithms in the Inverted Pendulum, Hopper, Humanoid, and Humanoid Standup environments, achieving up to 30% higher episodic rewards compared to the DPO algorithm. In the Walker2D and Half Cheetah environments, ResKPN also attains comparable results, while both the PPO and DPO algorithms struggle to perform well in these settings. The RKHS policy with representation networks takes more time to explore initially, leading to lower episodic returns at the start of training, as shown in the Inverted Pendulum, Hopper, and Humanoid environments. However, after sufficient exploration of the environment space, the RKHS policy achieves higher episodic rewards due to its sampling efficiency.

Considering the ablation results, it is significant that the AdvKPN and KPN fail to learn effective policies in complex environments such as Humanoid and Humanoid Standup. This reflects how the ResKPN algorithm leverages the advantages of ensemble learning with integrated representation networks. In contrast, the original RKHS method can only achieve similar performance to other

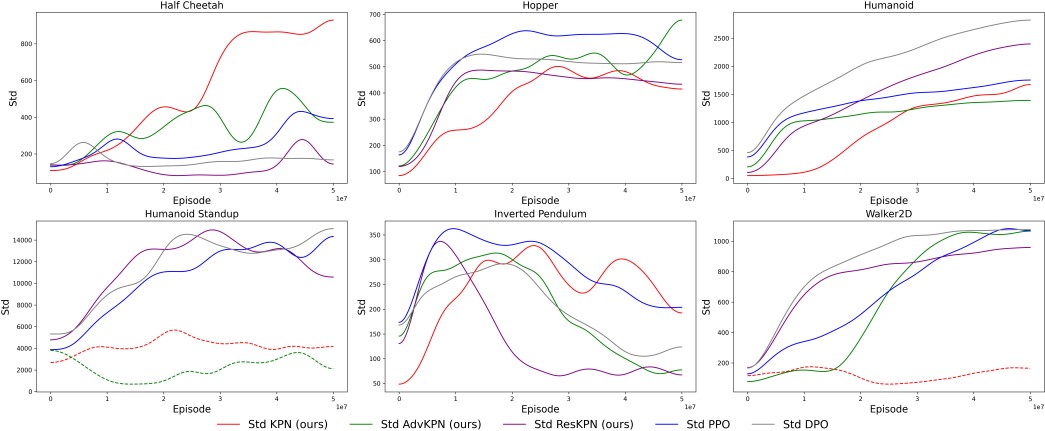

Figure 4: The standard deviation in MuJoCo environments for proposed algorithms.

algorithms in the simpler Inverted Pendulum environment, while it fails to learn effectively in more complex scenarios.

We also present the training variance in Figure 4. We observe that when all algorithms succeed in learning the local optimal policy in the Inverted Pendulum environment, the ResKPN algorithm exhibits the most stable training process overall. After integrating the advantage functions, the variance in AdvKPN decreases significantly compared to the KPN algorithm, which exhibits the highest variance similar to that of the PPO algorithm. For the insufficient learning observed in KPN and AdvKPN algorithms within complex environments, their variances in these situations are abnormally lower than those of algorithms that successfully learn a good policy, indicated by dashed lines in the figure. When considering the learned policies, it is observed that the ResKPN algorithm achieves relatively low variance, except in the Humanoid environment. More experiment comparisons are shown in Appendix D.

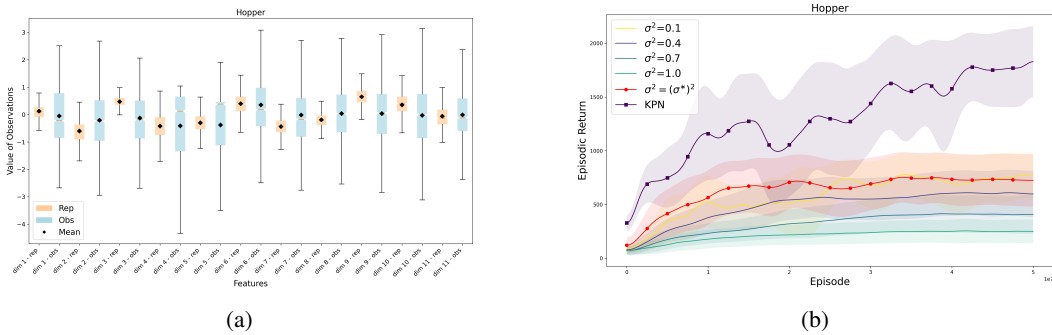

(a)                                                          (b)

Figure 5: The comparison of distributions and the learning performance after adjusting hyperparameters (a) The boxplot demonstrates the distribution differences of the observations (obs) and representations (rep) in different dimensions. (b) Comparison of episodic rewards for the RKHS policy with optimized hyperparameters versus alternative settings. The performance of the KPN algorithm is also illustrated.

### 4.3 ANALYSE FOR EFFECTIVENESS

In this section, we conduct a comprehensive evaluation addressing two key questions: (1) How effectively does representation learning adjust the observations of the original environment to align with the RKHS kernel? (2) Does the integration of advantage functions and the residual layer lead to a reduction in the variance of the RKHS gradient?

To answer (1), we consider the Gaussian kernel $K(x, y) = \exp\left(-\frac{\|x-y\|^2}{\sigma^2}\right)$, where the hyperparameter $\sigma$ can be viewed as an scaling of the variance of the observation distribution. The setting of the hyperparameter can imitate the scaling of observation embedding learned by the representation learning appaorch. In detail, let $\text{mean}(\text{var}_{obs})$ denote the mean variance of the original observations, and $\text{mean}(\text{var}_{rep})$ represent the mean variance of the representations. Then the controller $\sigma$ is set as $\sigma^* = \sqrt{\frac{\text{mean}(\text{var}_{obs})}{\text{mean}(\text{var}_{rep})}}$. The comparison of the distributions between observations and representations is illustrated in Figure 5a. It is shown that the representation networks tend to reduce the variance in the distribution of observations while maintaining a relatively unchanged mean. After tuning the hyperparameter, we observe a significant improvement in the learning performance of the RKHS policy, as shown in Figure 5b. The results indicate that the original RKHS policy achieves the best overall performance compared to other hyperparameter settings, although it still underperforms relative to the KPN, which dynamically processes observations through neural networks.

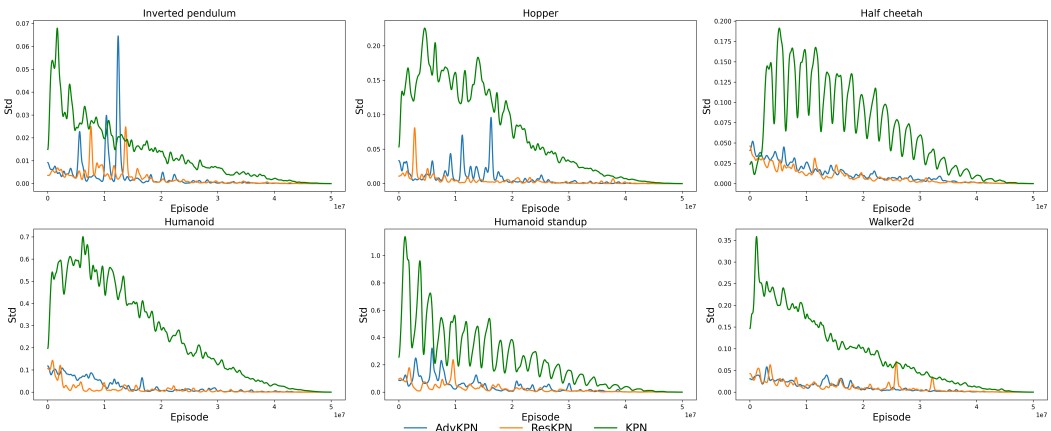

Figure 6: The estimated variance in MuJoCo environments for proposed algorithms.

To answer (2), we utilize a classic variance estimation formula, with detailed methodology provided in Appendix B.2. The variance of the RKHS gradient across various environments is illustrated in Figure 6.

The results indicate that the introduction of advantage functions significantly reduces the variance in KPN. Nevertheless, ephemeral fluctuations are still observed, particularly in the Inverted Pendulum, Hopper, and Humanoid Standup environments. The ResKPN further mitigates these fluctuations during training, demonstrating a rapid decrease in gradient variance at the beginning of the training process, notably in the Humanoid and Walker2D environments.

## 5 CONCLUSION

In this paper, we examine the limitations of RKHS policies, focusing on how inadequate representation learning and excessive variance cause unstable training and low robustness across environments. To address these issues, we introduce the ResKPN model, which capitalizes on the high sample efficiency of RKHS policies. Our theoretical analysis of ResKPN is validated, confirming the effectiveness of the proposed variance reduction methods. Comprehensive experiments demonstrate that our algorithm outperforms baseline methods.

However, several challenges remain. Unlike gradient updates in neural networks, which use minibatches for averaging gradients, RKHS gradient updates cannot currently leverage minibatches due to computational complexity. Additionally, the kernel's separate learning of high-dimensional spaces may hinder performance in multi-agent settings. Future research will focus on kernel embedding techniques to enable batch RKHS gradient updates and on approximating kernel computations to improve sample efficiency and reduce variance.

ACKNOWLEDGMENTS

We would like to acknowledge the support from the National Key R&D Program of China under Grant No. 2022ZD0160504 and the Shenzhen Key Laboratory of Ubiquitous Data Enabling under Grant No. ZDSYS20220527171406015.

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

# A  VARIANCE ANALYSE

## A.1  THE VARIANCE RATIO

In this section, we provide a proof to Lemma 3.1, which indicates that the variance ratio $\mathcal{R}$ satisfies:

$$\mathbb{E}_{s_k}\left[\frac{\mathrm{Var}_{a_k}(\nabla_h \hat{U}(\pi_h))}{\mathrm{Var}_{a_k}(\nabla_\theta \hat{U}(\pi_\theta))}\right] \geq \mathbb{E}_{s_k}\left[\frac{K^2(s_k, s_k)}{s_k^2}\frac{2c_1^2}{2c_2^2 + \Sigma^{-1}(c_2\theta^\top s_k + d_2)^2}\right].$$

First, we define the linear policy.

**Definition A.1 (Linear policy)** *The linear policy $\pi_\theta(a|s)$ is modeled as*

$$\pi_\theta(a|s) = \frac{1}{Z}e^{-\frac{1}{2}(\theta^\top s - a)^\top \mathbf{\Sigma}^{-1}(\theta^\top s - a)},$$

*where $\theta$ is the trainable parameter for policy. Linear policy embeds state $s$ in a linear manner as $\theta^\top s$.*

Comparing with RKHS policy in Equation (2), the only difference of lienar policy is the modeling of the mean vector $\theta^\top s$ and $h(s)$. Therefore, the estimated gradient of linear policy $\nabla_\theta \hat{U}(\pi_\theta)$ is derived (Zhao et al., 2011) as

$$\nabla_\theta \hat{U}(\pi_\theta) = \eta \cdot s_k \mathbf{\Sigma}^{-1}\left(a_k - \theta^\top s_k\right)\hat{Q}^{\pi_\theta}(a_k, s_k).$$

Then, we apply this definition in the variance ratio $\mathcal{R}$. Considering the expectation of variance ratio conditioned on $\vartheta$, combine with Eq. 3, we can have the variance ratio $\mathcal{R}$ as

$$\begin{aligned}
\mathcal{R} &= \frac{\mathrm{Var}_{a_k}(\nabla_h \hat{U}(\pi_h))}{\mathrm{Var}_{a_k}(\nabla_\theta \hat{U}(\pi_\theta))} = \frac{\eta^2 \, \mathrm{Var}_{a_k}(K(s_k, \cdot)\Sigma^{-1}(a_k - h(s_k))\hat{Q}^{\pi_h}(a_k, s_k))}{\eta^2 \, \mathrm{Var}_{a_k}(s_k \Sigma^{-1}\left(a_k - \theta^\top s_k\right)\hat{Q}^{\pi_\theta}(a_k, s_k))} \\
&= \frac{\langle K(s_k, \cdot), K(s_k, \cdot)\rangle \, \mathrm{Var}_{a_k}(K(s_k, \cdot)\Sigma^{-1}(a_k - h(s_k))\hat{Q}^{\pi_h}(a_k, s_k))}{s_k^2 \, \mathrm{Var}_{a_k}(\Sigma^{-1}\left(a_k - \theta^\top s_k\right)\hat{Q}^{\pi_\theta}(a_k, s_k))} \\
&= \frac{K(s_k, s_k) \, \mathrm{Var}_{a_k}(K(s_k, \cdot)\Sigma^{-1}(a_k - h(s_k))\hat{Q}^{\pi_h}(a_k, s_k))}{s_k^2 \, \mathrm{Var}_{a_k}(\Sigma^{-1}\left(a_k - \theta^\top s_k\right)\hat{Q}^{\pi_\theta}(a_k, s_k))},
\end{aligned}$$

where the last step uses the reproducing kernel property that $\forall x, y, \langle K(x, \cdot), K(y, \cdot)\rangle = K(x, y)$.

Considering the Assumption A.1 that $\hat{Q}^{\pi_\theta}(a_k, s_k)$ and $\hat{Q}^{\pi_h}(a_k, s_k)$ are linear functions conditioned on $s_k$, i.e., $\hat{Q}^{\pi_\theta}(a, s_k) = c_1^\top a + d_1$, $\hat{Q}^{\pi_h}(a, s_k) = c_2^\top a + d_2$, we can have that

$$\mathcal{R} = \frac{K(s_k, s_k) \, \mathrm{Var}_{a_k}(\Sigma^{-1}(a_k - h(s_k))(c_1 a_k + d_1))}{s_k^2 \, \mathrm{Var}_{a_k}(\Sigma^{-1}\left(a_k - \theta^\top s_k\right)(c_2 a_k + d_2))}.$$

We first consider the variance in the numerator part. Denote $y = \Sigma^{-\frac{1}{2}}(a_k - h(s_k))$, then we have that $y \sim \mathcal{N}(0, 1)$, then we can write $\mathrm{Var}\left(\Sigma^{-1}(a_k - h(s_k))(c_1 a_k + d_1)\right) = \mathrm{Var}\left(\Sigma^{-\frac{1}{2}}y(c_1 a_k + d_1)\right)$. Further, we have

$$\begin{aligned}
&\mathrm{Var}\left(\Sigma^{-\frac{1}{2}}y(c_1 a_k + d_1)\right) \\
&= \mathbb{E}\left[\left(\Sigma^{-\frac{1}{2}}y(c_1 a_k + d_1)\right)^2\right] - \left(\mathbb{E}\left[\Sigma^{-\frac{1}{2}}y(c_1 a_k + d_1)\right]\right)^2 \\
&= \Sigma^{-1}\mathbb{E}\left[\left(y(c_1(\Sigma^{\frac{1}{2}}y + h(s_k)) + d_1)\right)^2\right] - \left(\mathbb{E}\left[\Sigma^{-\frac{1}{2}}y(c_1(\Sigma^{\frac{1}{2}}y + h(s_k)) + d_1)\right]\right)^2 \\
&= \Sigma^{-1}\left[c_1^2 \Sigma \mathbb{E}[y^4] + 2c_1 \Sigma^{\frac{1}{2}}(c_1 h(s_k) + d_1)\mathbb{E}[y^3] + (c_1 h(s_k) + d_1)^2 \mathbb{E}[y^2]\right] \\
&\quad - \left(c_1 \mathbb{E}[y^2] + (c_1 h(s_k) + d_1)\mathbb{E}[y]\right)^2 \\
&= \Sigma^{-1}\left[c_1^2 \Sigma \cdot 3 + 2c_1 \Sigma^{\frac{1}{2}}(c_1 h(s_k) + d_1) \cdot 0 + (c_1 h(s_k) + d_1)^2 \cdot 1\right] \\
&\quad - \left(c_1 \cdot 1 + (c_1 h(s_k) + d_1) \cdot 0\right)^2 \\
&= 3c_1^2 + \Sigma^{-1}(c_1 h(s_k) + d_1)^2 - c_1^2 \\
&= 2c_1^2 + \Sigma^{-1}(c_1 h(s_k) + d_1)^2.
\end{aligned}$$

Using the similar method, the variance $\mathrm{Var}_{a_k}(\nabla_\theta \hat{U}(\pi_\theta))$ is derived as $2c_2^2 + \Sigma^{-1}(c_2\theta^\top s_k + d_2)^2$.

Therefore, the variance ratio is

$$\mathcal{R} = \frac{K^2(s_k, s_k)}{s_k^2} \frac{2c_1^2 + \Sigma^{-1}(c_1 h(s_k) + d_1)^2}{2c_2^2 + \Sigma^{-1}(c_2 s_k + d_2)^2} \geq \frac{K^2(s_k, s_k)}{s_k^2} \frac{2c_1^2}{2c_2^2 + \Sigma^{-1}(c_2\theta^\top s_k + d_2)^2}.$$

In this case,

$$\mathbb{E}_{s_k}\left[\frac{\mathrm{Var}_{a_k}(\nabla_h \hat{U}(\pi_h))}{\mathrm{Var}_{a_k}\nabla_\theta \hat{U}(\pi_\theta))}\right] \geq \mathbb{E}_{s_k}\left[\frac{K^2(s_k, s_k)}{s_k^2} \frac{2c_1^2}{2c_2^2 + \Sigma^{-1}(c_2\theta^\top s_k + d_2)^2}\right].$$

## A.2 THE VARIANCE IN DIFFERENT RKHS GRADIENT ALGORITHMS

In this section, we prove the Theorem 3.2.

**Advantage Function Effect** First, we prove that $\mathrm{Var}_{a_k}(\nabla_h \hat{U}(\pi_\varpi)) \geq \mathrm{Var}_{a_k}(\nabla_h \hat{U}_{A^{\pi_\varpi}}(\pi_\varpi))$.

For simplification of notations, we denote $\mathrm{Var}_{a_k}(\nabla_h \hat{U}_{A^{\pi_\varpi}}(\pi_\varpi)) = \mathbb{E}_{a_k}\left[(\Gamma_A - \mathbb{E}_{a_k}[\Gamma_A])^2\right]$ with

$$\Gamma_A = \nabla_h \hat{U}_{A^{\pi_\varpi}}(\pi_\varpi) = \nabla \log \pi_\varpi(a_k|s_k) A^{\pi_\varpi},$$

and $\mathrm{Var}_{a_k}(\nabla_h \hat{U}(\pi_\varpi)) = \mathbb{E}_{a_k}[(\Gamma_Q - \mathbb{E}_{a_k}[\Gamma_Q])^2]$ with

$$\Gamma_Q = \nabla_h \hat{U}(\pi_\varpi) = \nabla \log \pi_\varpi(a_k|s_k) Q^{\pi_\varpi}(a_k, s_k).$$

Similarly, we can denote that $\Gamma_V = \mathbb{E}_{a_k}[(\Gamma_V)^2]$ with $\nabla \log \pi_\varpi(a_k|s_k) V^{\pi_\varpi}(s_k)$. Since $A^{\pi_\varpi} = Q^{\pi_\varpi}(a_k, s_k) - V^{\pi_\varpi}(s_k)$, we can derive that:

$$\Gamma_A - \mathbb{E}_{a_k}[\Gamma_A]$$
$$= \nabla \log \pi_\varpi(a_k|s_k) A^{\pi_\varpi} - \mathbb{E}_{a_k}[\nabla \log \pi_\varpi(a_k|s_k) A^{\pi_\varpi}]$$
$$= \nabla \log \pi_\varpi(a_k|s_k)(Q^{\pi_\varpi}(a_k, s_k) - V^{\pi_\varpi}(s_k)) - \mathbb{E}_{a_k}[\nabla \log \pi_\varpi(a_k|s_k)(Q^{\pi_\varpi}(a_k, s_k) - V^{\pi_\varpi}(s_k))]$$
$$= \left(\nabla \log \pi_\varpi(a_k|s_k) Q^{\pi_\varpi}(a_k, s_k) - \mathbb{E}_{a_k}[\nabla \log \pi_\varpi(a_k|s_k) Q^{\pi_\varpi}(a_k, s_k)]\right) -$$
$$\quad \left(\nabla \log \pi_\varpi(a_k|s_k) V^{\pi_\varpi}(s_k) - \mathbb{E}_{a_k}[\nabla \log \pi_\varpi(a_k|s_k) V^{\pi_\varpi}(s_k)]\right)$$
$$= (\Gamma_Q - \mathbb{E}_{a_k}[\Gamma_Q]) - (\Gamma_V - \mathbb{E}_{a_k}[\Gamma_V]).$$

Therefore, we have that

$$\mathrm{Var}_{a_k}(\nabla_h \hat{U}_{A^{\pi_\varpi}}(\pi_\varpi))$$
$$= \mathbb{E}_{a_k}\left[(\Gamma_A - \mathbb{E}_{a_k}[\Gamma_A])^2\right]$$
$$= \mathbb{E}_{a_k}\left[\left((\Gamma_Q - \mathbb{E}_{a_k}[\Gamma_Q]) - (\Gamma_V - \mathbb{E}_{a_k}[\Gamma_V])\right)^2\right]$$
$$= \mathbb{E}_{a_k}\left[(\Gamma_Q - \mathbb{E}_{a_k}[\Gamma_Q])^2 - 2(\Gamma_Q - \mathbb{E}_{a_k}[\Gamma_Q])(\Gamma_V - \mathbb{E}_{a_k}[\Gamma_V]) + (\Gamma_V - \mathbb{E}_{a_k}[\Gamma_V])^2\right]$$
$$= \mathrm{Var}_{a_k}(\nabla_h \hat{U}(\pi_\varpi)) + \mathbb{E}_{a_k}\left[-2(\Gamma_Q - \mathbb{E}_{a_k}[\Gamma_Q])(\Gamma_V) + (\Gamma_V)^2\right]$$
$$= \mathrm{Var}_{a_k}(\nabla_h \hat{U}(\pi_\varpi)) + \mathbb{E}_{a_k}\left[(\Gamma_V)^2 - 2\Gamma_Q\Gamma_V\right]$$
$$= \mathrm{Var}_{a_k}(\nabla_h \hat{U}(\pi_\varpi)) + (V^{\pi_\varpi})^2(s_k)\mathbb{E}_{a_k}[\langle \nabla \log \pi_\varpi(a_k|s_k), \nabla \log \pi_\varpi(a_k|s_k)\rangle] -$$
$$\quad 2V^{\pi_\varpi}(s_k)\mathbb{E}_{a_k}[\langle \nabla \log \pi_\varpi(a_k|s_k) Q^{\pi_\varpi}(a_k, s_k), \nabla \log \pi_\varpi(a_k|s_k)\rangle],$$

where we use the $\mathbb{E}_{a_k}[\nabla \log \pi_\varpi(a_k|s_k) V^{\pi_\varpi}(s_k)] = 0$ property for $\nabla \log \pi_\varpi(a_k|s_k)$ (Weaver & Tao, 2001). Therefore, it is easily derived that

$$\Gamma_Q^2 - \Gamma_A^2 = -(V^{\pi_\varpi})^2 \mathbb{E}_{a_k}[\langle \nabla \log \pi_\varpi(a_k|s_k), \nabla \log \pi_\varpi(a_k|s_k)\rangle] +$$
$$\quad 2V^{\pi_\varpi}(s_k)\mathbb{E}_{a_k}[\langle \nabla \log \pi_\varpi(a_k|s_k) Q^{\pi_\varpi}(a_k, s_k), \nabla \log \pi_\varpi(a_k|s_k)\rangle]$$
$$\geq 0,$$

where the last inequality uses the assumption $Q^{\pi_\varpi}(a_k, s_k) \geq \frac{1}{2}\mathbb{E}_{a_k}[Q^{\pi_\varpi}(a_k, s_k)] = \frac{1}{2}V^{\pi_\varpi}(s_k)$. Therefore, it is shown that $\mathrm{Var}_{a_k}(\nabla_h \hat{U}(\pi_\varpi)) \geq \mathrm{Var}_{a_k}(\nabla_h \hat{U}_{A^{\pi_\varpi}}(\pi_\varpi))$.

**Residual Effect** Now we consider the variance of $\nabla_h \hat{U}_{A^{\pi_\varpi}}(\pi_{\varpi,\iota})$. Similarly, we write $\mathrm{Var}_{a_k}(\nabla_h \hat{U}_{A^{\pi_\varpi}}(\pi_{\varpi,\iota})) = \mathbb{E}_{a_k}\left[(\Gamma_{h,\mu} - \mathbb{E}_{a_k}[\Gamma_{h,\mu}])^2\right]$, where $\Gamma_{h,\mu} = \nabla_h \hat{U}_{A^{\pi_\varpi}}(\pi_{\varpi,\iota}) = \mathrm{K}(s_k,\cdot)\Sigma^{-1}(a_k - h(\psi_\vartheta(s_k)) - \mu_\iota(\psi_\vartheta(s_k)))A^{\pi_\varpi}(a_k,s_k)$. Additionally, we denote that $\mathrm{Var}_{a_k}(\nabla_h \hat{U}_{A^{\pi_\varpi}}(\pi_\varpi)) = \mathbb{E}_{a_k}\left[(\Gamma_h - \mathbb{E}_{a_k}[\Gamma_h])^2\right]$ with $\Gamma_h = \mathrm{K}(s_k,\cdot)\Sigma^{-1}(a_k - h(\psi_\vartheta(s_k)))A^{\pi_\varpi}(a_k,s_k)$ and $\Gamma_\mu = \mathrm{K}(s_k,\cdot)\Sigma^{-1}(\mu_\iota(\psi_\vartheta(s_k)))A^{\pi_\varpi}(a_k,s_k)$. Like the derivations in the advantage function effect, it is easily observed that

$$\Gamma_{h,\mu} - \mathbb{E}_{a_k}[\Gamma_{h,\mu}] = (\Gamma_h - \mathbb{E}_{a_k}[\Gamma_h]) - (\Gamma_\mu - \mathbb{E}_{a_k}[\Gamma_\mu]).$$

Based on the definition, we have that

$$
\begin{aligned}
&\mathrm{Var}_{a_k}(\nabla_h \hat{U}_{A^{\pi_\varpi}}(\pi_{\varpi,\iota}))\\
=&\mathbb{E}_{a_k}\left[(\Gamma_{h,\mu} - \mathbb{E}_{a_k}[\Gamma_{h,\mu}])^2\right]\\
=&\mathbb{E}_{a_k}\left[\left((\Gamma_h - \mathbb{E}_{a_k}[\Gamma_h]) - (\Gamma_\mu - \mathbb{E}_{a_k}[\Gamma_\mu])\right)^2\right]\\
=&\mathrm{Var}_{a_k}(\nabla_h \hat{U}_{A^{\pi_\varpi}}(\pi_\varpi)) + \mathbb{E}_{a_k}\left[-2(\Gamma_h - \mathbb{E}_{a_k}[\Gamma_h])(\Gamma_\mu) + (\Gamma_\mu)^2\right]\\
=&\Gamma_A^2 + \mathbb{E}_{a_k}\left[\langle -2\mathrm{K}(s_k,\cdot)\Sigma^{-1}(a_k - h(\psi_\vartheta(s_k)))A^{\pi_\varpi}(a_k,s_k), \mathrm{K}(s_k,\cdot)\Sigma^{-1}\mu_\iota(\psi_\vartheta(s_k))A^{\pi_\varpi}(a_k,s_k)\rangle\right.\\
&\left.+\langle \mathrm{K}(s_k,\cdot)\Sigma^{-1}\mu_\iota(\psi_\vartheta(s_k))A^{\pi_\varpi}(a_k,s_k), \mathrm{K}(s_k,\cdot)\Sigma^{-1}\mu_\iota(\psi_\vartheta(s_k))A^{\pi_\varpi}(a_k,s_k)\rangle\right],
\end{aligned}
$$

where we use the property $\mathbb{E}_{a_k}\left[\mathrm{K}(s_k,\cdot)\Sigma^{-1}\mu_\iota(\psi_\vartheta(s_k))A^{\pi_\varpi}(a_k,s_k)\right] = 0$ due to $\mathbb{E}_{a_k}[A^{\pi_\varpi}(a_k,s_k)] = 0$, then we can derive that

$$
\begin{aligned}
&\mathrm{Var}_{a_k}(\nabla_h \hat{U}_{A^{\pi_\varpi}}(\pi_\varpi)) - \mathrm{Var}_{a_k}(\nabla_h \hat{U}_{A^{\pi_\varpi}}(\pi_{\varpi,\iota}))\\
=&\mathbb{E}_{a_k}\left[\langle 2\mathrm{K}(s_k,\cdot)\Sigma^{-1}(a_k - h(\psi_\vartheta(s_k)))A^{\pi_\varpi}(a_k,s_k), \mathrm{K}(s_k,\cdot)\Sigma^{-1}\mu_\iota(\psi_\vartheta(s_k))A^{\pi_\varpi}(a_k,s_k)\rangle\right.\\
&\left.-\langle \mathrm{K}(s_k,\cdot)\Sigma^{-1}\mu_\iota(\psi_\vartheta(s_k))A^{\pi_\varpi}(a_k,s_k), \mathrm{K}(s_k,\cdot)\Sigma^{-1}\mu_\iota(\psi_\vartheta(s_k))A^{\pi_\varpi}(a_k,s_k)\rangle\right]\\
\geq&\mathbb{E}_{a_k}\left[\langle \mathrm{K}(s_k,\cdot)\Sigma^{-1}\mu_\iota(\psi_\vartheta(s_k))A^{\pi_\varpi}(a_k,s_k), \mathrm{K}(s_k,\cdot)\Sigma^{-1}\mu_\iota(\psi_\vartheta(s_k))A^{\pi_\varpi}(a_k,s_k)\rangle\right.\\
&\left.-\langle \mathrm{K}(s_k,\cdot)\Sigma^{-1}\mu_\iota(\psi_\vartheta(s_k))A^{\pi_\varpi}(a_k,s_k), \mathrm{K}(s_k,\cdot)\Sigma^{-1}\mu_\iota(\psi_\vartheta(s_k))A^{\pi_\varpi}(a_k,s_k)\rangle\right]\\
\geq&0
\end{aligned}
$$

Therefore we can attain that $\mathrm{Var}_{a_k}(\nabla_h \hat{U}(\pi_\varpi)) \geq \mathrm{Var}_{a_k}(\nabla_h \hat{U}_{A^{\pi_\varpi}}(\pi_\varpi)) \geq \mathrm{Var}_{a_k}(\nabla_h \hat{U}_{A^{\pi_\varpi}}(\pi_{\varpi,\iota}))$, where the last inequality uses the assumption $a_k - h(\psi_\vartheta(s_k)) \geq \frac{1}{2}\mu_\iota(\psi_\vartheta(s_k))$.

## B  DETAILS IN EXPERIMENT

The experimental details are shown in this section, including a scheme figure of the ResKPN algorithm, the techniques in PPO and DPO integrated with the proposed algorithms, the function to estimate the variance of the RKHS gradient, and the hyperparameters in the experiments.

Without loss of generality, we adopt the Gaussian kernel for the experiments due to its well-established properties in capturing smooth and continuous relationships in high-dimensional spaces. To leverage GPU acceleration for training and simulation, we utilize the Brax simulator, which is developed using Jax (Freeman et al., 2021). The experiments are conducted on a cluster server, with each experiment utilizing an NVIDIA RTX A6000 GPU and 32 cores of an Intel(R) Xeon(R) Gold 5218 CPU running at 2.30 GHz.

### B.1  THE IMPLEMENTATION DETAILS IN RESKPN

The methodology scheme of ResKPN is shown in Figure 7, where $\nabla_\iota \hat{U}(\pi_{\varpi,\iota})$ represents the gradient in the residual layer $\mu_\iota(\psi_\vartheta(\cdot))$, and the notation FC means the Fully Connected layer, which composes the representation networks. The observations are initially input into the representation networks for distribution adjustment. Both RKHS function $h(\psi_\vartheta(s_k))$ and residual layer

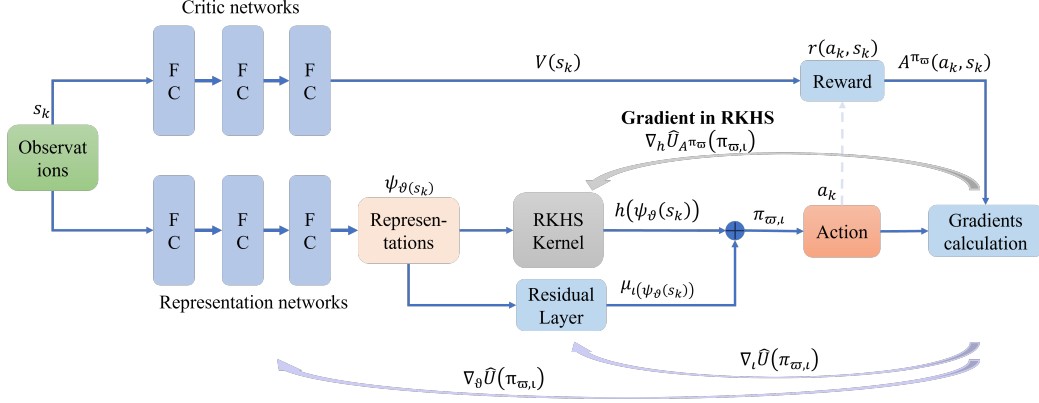

Figure 7: The scheme of the ResKPN algorithm.

$\mu_\iota(\psi_\vartheta(s_k)))$ uses the representations for action choosing. Additionally, the covariance matrix $\mathbf{\Sigma}^{-1}$ is parameterized as the network parameters, which are updated through the backpropagation. The main difference between ResKPN and AdvKPN algorithm is the usage of the residual layer, stabilizing the training process.

To leverage the classic variance reduction techniques, We employ the techniques of Proximal Policy Optimization (PPO) algorithms in the KPN, AdvKPN, and ResKPN algorithms to reduce variance. Specifically, the optimization objective in PPO utilizes Trust Region Policy Optimization (TRPO) (Schulman et al., 2015), which we adapt to optimize our gradient. The optimization objective in TRPO is given by

$$
\begin{aligned}
\underset{\theta}{\text{maximize}} \quad & \hat{\mathbb{E}}_k \left[ \frac{\pi_\theta(a_k \mid s_k)}{\pi_{\theta_{\text{old}}}(a_k \mid s_k)} \hat{A}_k \right] \\
\text{subject to} \quad & \hat{\mathbb{E}}_k \left[ \text{KL} \left[ \pi_{\theta_{\text{old}}}(\cdot \mid s_k), \pi_\theta(\cdot \mid s_k) \right] \right] \leq \delta,
\end{aligned}
$$

where $\theta_{\text{old}}$ represents the parameters before the update. Inspired by this formulation, since

$$
\frac{\nabla \pi_h(a_k \mid s_k)}{\pi_{h_{\text{old}}}(a_k \mid s_k)} = \frac{\pi_h(a_k \mid s_k)}{\pi_{h_{\text{old}}}(a_k \mid s_k)} \nabla \log \pi_h(a_k \mid s_k),
$$

we adapt the RKHS policy gradient by scaling the original gradient with the policy ratio $\frac{\pi_h(a_k \mid s_k)}{\pi_{h_{\text{old}}}(a_k \mid s_k)}$. Additionally, we incorporate the clipping technique used in PPO, where the policy ratio is clipped as follows:

$$
\min \left( \frac{\pi_h(a_k \mid s_k)}{\pi_{h_{\text{old}}}(a_k \mid s_k)}, \text{clip} \left( \frac{\pi_h(a_k \mid s_k)}{\pi_{h_{\text{old}}}(a_k \mid s_k)}, 1 - \epsilon, 1 + \epsilon \right) \right).
$$

The threshold value $\epsilon$ is set to $0.2$ in the PPO algorithm. In contrast, we set $\epsilon$ to $1$ in the RKHS policy to encourage more exploration, leveraging the high sample efficiency of the RKHS gradient. To balance the exploration-exploitation trade-off, we introduce the drift function $\mathcal{D}$ from Discovered Policy Optimization (DPO). The main difference between DPO and the PPO algorithm is the inclusion of a drift function that depends on the advantage function values, allowing the adjustment of hyperparameters to control the degree of exploration and exploitation in the policy. The drift function is calculated as:

$$\mathcal{D} = \begin{cases} \max\left(0, \left(\frac{\pi_h(a_k \mid s_k)}{\pi_{h_{\text{old}}}(a_k \mid s_k)} - 1\right) A^{\pi_\varpi} - \alpha_{\mathcal{D}} \tanh\left(\frac{\left(\frac{\pi_h(a_k \mid s_k)}{\pi_{h_{\text{old}}}(a_k \mid s_k)} - 1\right) A^{\pi_\varpi}}{\alpha_{\mathcal{D}}}\right)\right), & \text{if } A^{\pi_\varpi} \geq 0, \\[2em] \max\left(0, \frac{\pi_h(a_k \mid s_k)}{\pi_{h_{\text{old}}}(a_k \mid s_k)} A^{\pi_\varpi} - \beta_{\mathcal{D}} \tanh\left(\frac{\frac{\pi_h(a_k \mid s_k)}{\pi_{h_{\text{old}}}(a_k \mid s_k)} A^{\pi_\varpi}}{\beta_{\mathcal{D}}}\right)\right), & \text{otherwise,} \end{cases}$$

where the hyperparameters $\alpha_{\mathcal{D}}$ and $\beta_{\mathcal{D}}$ balance the exploration and exploitation of the policy. The drift function $\mathcal{D}$ is integrated into the RKHS gradient as:

$$\nabla_h \hat{U}_{A^{\pi_\varpi}}(\pi_{\varpi,\iota}) = \eta K(\psi_\vartheta(s_k), \cdot) \Sigma^{-1} \left(a_k - h(\psi_\vartheta(s_k)) - \mu_\iota(\psi_\vartheta(s_k))\right) \left(A^{\pi_\varpi}(a_k, s_k) - \mathcal{D}\right).$$

Table 1: Comparison of Hyperparameters among ResKPN, PPO, and DPO.

| Parameter | ResKPN | PPO | DPO |
|---|---|---|---|
| **Optimization** | | | |
| Learning Rate | $3 \times 10^{-4}$ | $3 \times 10^{-4}$ | $3 \times 10^{-4}$ |
| Learning Rate for RKHS | $1 \times 10^{-1}$ | N/A | N/A |
| Adam for RKHS | 0.9 | N/A | N/A |
| Max Gradient Norm | 0.5 | 0.5 | 0.5 |
| **Environment** | | | |
| Number of Environments in Parallel | 2048 | 2048 | 2048 |
| **Training Schedule** | | | |
| Number of Steps per Update | 10 | 10 | 10 |
| Total Timesteps | $5 \times 10^7$ | $5 \times 10^7$ | $5 \times 10^7$ |
| Number of Minibatches | 32 | 32 | 32 |
| **Algorithm Specific** | | | |
| GAE Lambda | 0.95 | 0.95 | 0.95 |
| Clip Epsilon | 0.2 | 0.2 | 0.2 |
| Clip Epsilon for RKHS | 1 | N/A | N/A |
| DPO Alpha | 2.0 | N/A | 2.0 |
| DPO Beta | 0.6 | N/A | 0.6 |
| DPO Alpha RKHS | 1 | N/A | N/A |
| DPO Beta RKHS | 0.6 | N/A | N/A |
| **Model Architecture** | | | |
| Feature Dimension | 256 | N/A | N/A |
| Activation Function | tanh | tanh | tanh |

## B.2 THE ESTIMATION OF VARIANCE IN THE RKHS GRADIENT

To estimate the variance of RKHS gradient in the experiment, we conduct $\mathcal{G}$ parallel experiments. It is observed that the RKHS gradient can be denoted as $\alpha_t K(s_t, \cdot)$, where $\alpha_t$ represents the coefficient including the learning rate, the derivative of log policy and the $Q$-function value or advantage function, and the kernel is determined by the observations $s_t$. Therefore, we can derive the following

variance estimation equation:

$$\widehat{\text{Var}}_{a_k,s_k}(\nabla_\theta \hat{U}(\pi_h)) = \frac{1}{\mathcal{G}} \sum_{t=1}^{\mathcal{G}} \langle \alpha_t K(s_t, \cdot) - \frac{1}{\mathcal{G}} \sum_{i=1}^{\mathcal{G}} \alpha_i K(s_i, \cdot), \alpha_t K(s_t, \cdot) - \frac{1}{\mathcal{G}} \sum_{i=1}^{\mathcal{G}} \alpha_i K(s_i, \cdot) \rangle$$

$$= \frac{1}{\mathcal{G}} \sum_{t=1}^{\mathcal{G}} \left[ \alpha_t K(s_t, s_t) \alpha_t^\top - 2\alpha_t \frac{1}{\mathcal{G}} \sum_{i=1}^{\mathcal{G}} K(s_i, s_t) \alpha_i^\top + \frac{1}{\mathcal{G}^2} \sum_{i=1}^{\mathcal{G}} \sum_{j=1}^{\mathcal{G}} \alpha_i K(s_i, s_j) \alpha_j^\top \right],$$

$$(10)$$

where we use the reproducing property of the RKHS kernel.

### B.3 HYPERPARAMETERS IN THE EXPERIMENT

The hyperparameters for PPO, DPO and ResKPN are shown in Table 1. The Max Gradient Norm only limits the gradient update of the representation networks in ResKPN. We run environments in parallel based on the JAX, which supports the acceleration of simulation in GPUs. All experiments run for $5 \times 10^7$ steps to compare the learning speed and convergence performance.

## C   MORE EXPLANATIONS FOR THE VARIANCE REDUCTION EFFECT IN THE RESIDUAL LAYER

One of the main reasons that the advantage function $A^{\pi_\varpi}(a_k, s_k) = Q^{\pi_\varpi}(a_k, s_k) - V^{\pi_\varpi}(s_k)$ can reduce the variance of the gradient is the natural properties that $V^{\pi_\varpi}(s_k) = \mathbb{E}_{a_k}[Q^{\pi_\varpi}(a_k, s_k)]$, which indicates that the state value function $V^{\pi_\varpi}(s_k)$ is potentially stable than the Q-function $Q^{\pi_\varpi}(a_k, s_k)$ (Greensmith et al., 2004). We compare these three values in the Hopper environment, which is shown in Figure 8. It can be seen that after subtracting the state value function $V^{\pi_\varpi}(s_k)$, the advantage function shows more stability and a low-value curve, which avoids the overestimation (Greensmith et al., 2004).

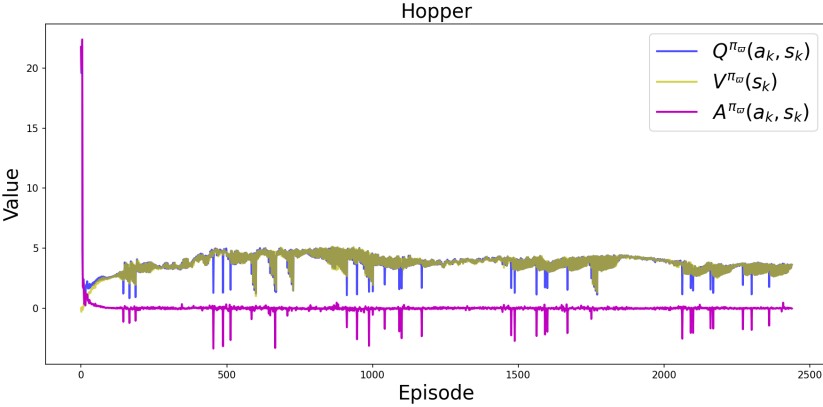

Figure 8: The variance reduction by introducing the advantage function in Hopper environment.

Inspired by the baseline function in the advantage framework, we introduce the residual layer $\mu_\iota(\cdot)$ as a "baseline function" within the RKHS policy. This addition significantly enhances training stability compared to the RKHS function $h(\cdot)$. To demonstrate the variance reduction achieved by the residual layer, we compare the values of $a_k - h(\psi_\vartheta(s_k)) - \mu_\iota(\psi_\vartheta(s_k))$, $a_k - h(\psi_\vartheta(s_k))$, and $\mu_\iota(\psi_\vartheta(s_k))$ in Figures 9 and 10, across the Inverted Pendulum and Hopper environments. These comparisons reveal that introducing the residual layer as a "baseline function" stabilizes the product term $(a_k - h(\psi_\vartheta(s_k)) - \mu_\iota(\psi_\vartheta(s_k)))$ relative to the original RKHS term $(a_k - h(\psi_\vartheta(s_k)))$. This stabilization directly contributes to the variance reduction in the RKHS gradient:

$$\nabla_h \hat{U}_{A^{\pi_\varpi}}(\pi_{\varpi,\iota}) = \eta K(\psi_\vartheta(s_k), \cdot) \Sigma^{-1}(a_k - h(\psi_\vartheta(s_k)) - \mu_\iota(\psi_\vartheta(s_k))) A^{\pi_\varpi}(a_k, s_k).$$

This also provides additional evidence supporting the correctness of Theorem 3.2.

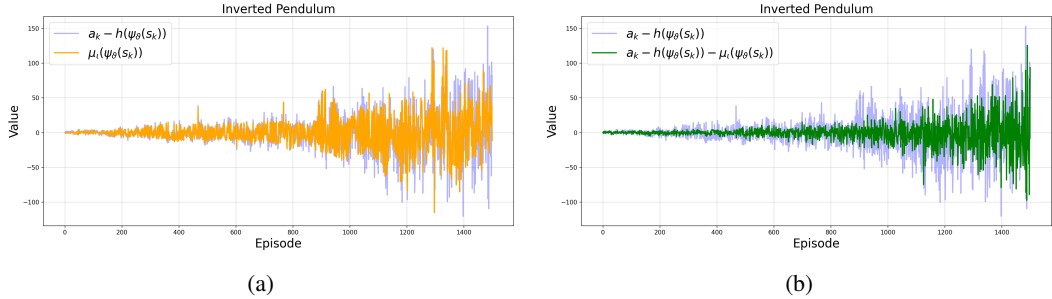

(a)                                          (b)

Figure 9: Variance reduction achieved by introducing the residual layer $\mu_\iota(\cdot)$ in the Inverted Pendulum environment. (a) Comparison of $a_k - h(\psi_\vartheta(s_k))$ and $\mu_\iota(\psi_\vartheta(s_k))$. (b) Comparison of $a_k - h(\psi_\vartheta(s_k))$ and $a_k - h(\psi_\vartheta(s_k)) - \mu_\iota(\psi_\vartheta(s_k))$.

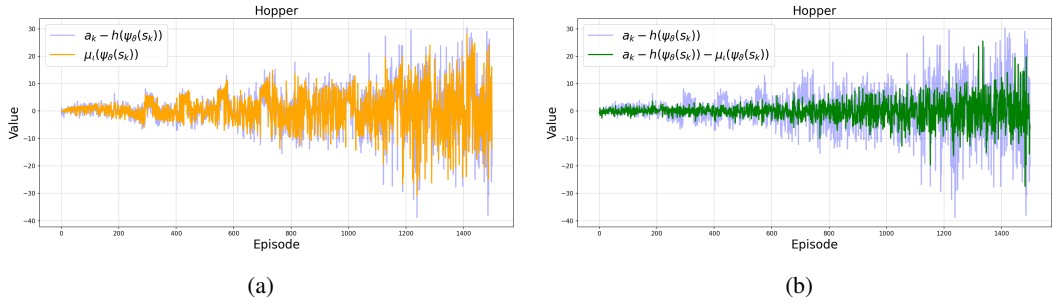

(a)                                          (b)

Figure 10: Variance reduction achieved by introducing the residual layer $\mu_\iota(\cdot)$ in the Hopper environment. (a) Comparison of $a_k - h(\psi_\vartheta(s_k))$ and $\mu_\iota(\psi_\vartheta(s_k))$. (b) Comparison of $a_k - h(\psi_\vartheta(s_k))$ and $a_k - h(\psi_\vartheta(s_k)) - \mu_\iota(\psi_\vartheta(s_k))$.

## D    MORE EXPERIMENT COMPARISONS

### D.1    MORE VARIANCE REDUCTION METHODS COMPARISON

In this section, we delve into variance reduction techniques from related literature and present an experiment on the minibatch method to highlight the distinctions and advantages of our approach compared to prior research.

Table 2: Variance Reduction Methods for RKHS Policies.

| Paper | Variance Reduction Method | Tested Environment | State | Action |
|---|---|---|---|---|
| (Le et al., 2019) | Learning rate search / Kernel matching pursuit | Quadrotor Navigation | 13 | 3 |
| (Paternain et al., 2020) | Symmetric estimation | Self-Charging Surveillance Robot | 3 | 2 |
| (Mazoure et al., 2020) | Predefined kernel orthogonal basis | Pendulum | 3 | 1 |
| (Wang & Principe, 2021) | Kernel orthogonal bases clustering | Nonhuman primate performing an obstacle-avoidance task | / | 1 |

Table 2 summarizes various variance reduction methods for RKHS-based reinforcement learning policies, highlighting their tested environments, techniques, and applicable state-action dimensions. These approaches, while effective in specific scenarios, face significant challenges when applied to high-dimensional environments like MuJoCo. For instance, the model-based method in (Paternain et al., 2020) reduces gradient variance by averaging gradients of symmetric transitions, a strategy

that is effective in simple tasks such as self-charging surveillance robots but struggles in complex settings where symmetric transitions are difficult to identify. Similarly, (Le et al., 2019) introduces a kernel matching pursuit method that reduces variance through gradient regression, yet it suffers from computational instability in tasks with large state-action spaces, such as those found in MuJoCo environments.

The method in (Mazoure et al., 2020) proposes truncating RKHS embeddings to represent policies using kernel orthogonal bases, theoretically reducing Q-function variance. However, its reliance on partitioning the state space into fixed bins renders it impractical for high-dimensional environments. For example, in the Hopper environment with a 17-dimensional state space, even simple binary division across dimensions leads to an exponential number of kernel bases, making the approach computationally infeasible. Similarly, (Wang & Principe, 2021) employs an online clustering method to aggregate kernel orthogonal bases, reducing computational costs and variance in neural signal processing tasks. While this technique offers valuable insights, its focus on Q-function learning rather than policy gradient optimization, coupled with the bias introduced by clustering, limits its applicability to RKHS policy frameworks in high-dimensional settings. These methods collectively underscore the challenges of applying variance reduction techniques effectively in complex environments like MuJoCo, where computational efficiency and scalability remain critical obstacles.

Although variance reduction techniques have been specifically designed for RKHS gradients, minibatch gradient computation (Qian & Klabjan, 2020) is widely adopted in deep learning and many machine learning tasks due to its effectiveness in stabilizing gradient updates. However, applying minibatch gradients directly to RKHS methods introduces significant computational challenges. The mean of RKHS gradients must be explicitly expressed as $\sum_{i=1}^{n} \alpha_i K(s_i, \cdot)$, where $n$ represents the minibatch size. This results in a quadratic increase in computational complexity as $n$ grows, making minibatch gradients computationally prohibitive for RKHS-based methods.

To evaluate the impact of minibatch size on computational cost, we conducted experiments with varying minibatch sizes in two environments, and the results are summarized in Table 3.

Table 3: Training Time for Different Minibatch Sizes Across Environments (in minutes).

| Environment | Training Time (min.)/ Minibatch Size $n$ | | | | |
|---|---|---|---|---|---|
| | $n=1$ | $n=2$ | $n=3$ | $n=4$ | $n=5$ |
| Half Cheetah | 13.19 | 61.25 | 140.74 | 199.49 | 205.07 |
| Hopper | 10.83 | 58.91 | 123.12 | 169.03 | 202.61 |

The results demonstrate that training time increases quadratically with minibatch size in both the Hopper and Half Cheetah environments. For example, using a minibatch size of $n=5$ requires over three additional hours of computation compared to $n=1$. These findings underscore the computational infeasibility of employing minibatch gradients in RKHS-based methods for high-dimensional environments. Despite the significant computational cost, using minibatches remains important due to their ability to reduce training variance. We conducted experiments with a minibatch size of $n=2$ (as larger sizes are prohibitively slow) to evaluate training performance and variance, as shown in Figures 11 and 12. These figures indicate that the training performance with $n=2$ minibatches shows minimal differences compared to the original approach. However, incorporating an additional transition per training period effectively reduces the variance of both AdvKPN and ResKPN, highlighting the potential of the minibatch approach.

If the computational cost associated with minibatches can be addressed, we believe that this design has the potential to further reduce variance in RKHS policy methods, offering a promising direction for future research.

## D.2 EXPERIMENTS ON OTHER ENVIRONMENTS

In this section, we extend the experimental evaluation of our proposed algorithms to two additional environments: Pusher and Reacher. The episodic rewards obtained during training are presented in Figure 13.

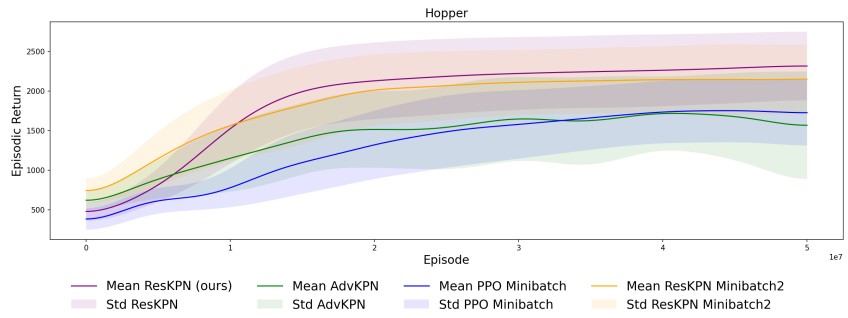

Figure 11: The episodic reward in Hopper environment for minibatch size $n = 2$.

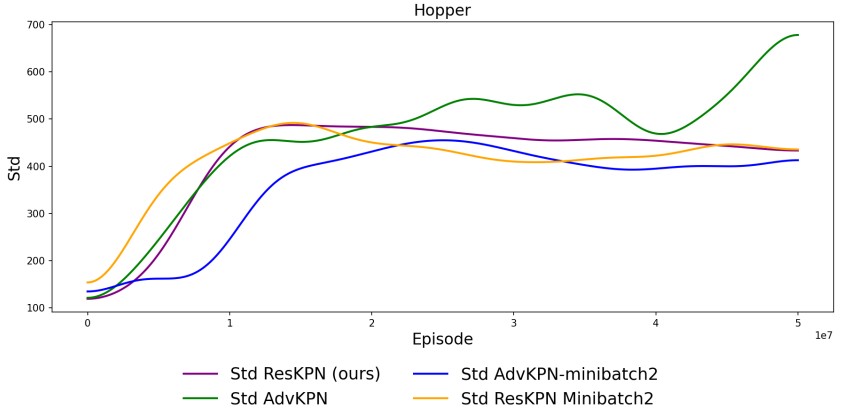

Figure 12: The standard deviation in Hopper environment for minibatch size $n = 2$.

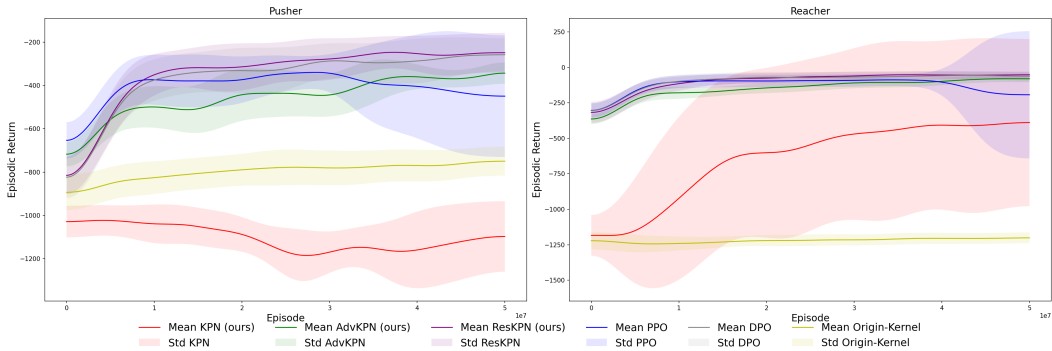

Figure 13: The episodic reward in Pusher and Reacher environments for proposed algorithms.

It is illustrated in the Figure 13 that the ResKPN consistently achieves the best performance in the Pusher environment. In the Reacher environment, all algorithms except KPN and Origin-Kernel converge to the optimal reward. This demonstrates the robustness of ResKPN, particularly in complex environments requiring precise control. To further analyze stability during training, the standard deviations of the episodic rewards for the proposed algorithms are shown in Figure 14. From Fig-

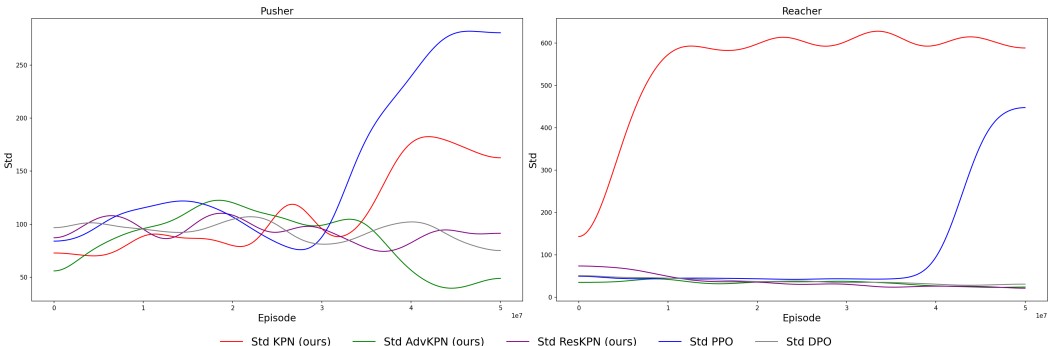

Figure 14: The standard deviation in Pusher and Reacher environments for proposed algorithms.

ure 14, it is evident that ResKPN, DPO, and AdvKPN achieve the smallest overall variance in the Pusher environment, indicating stable learning dynamics. In the Reacher environment, while all algorithms (except KPN) converge, the PPO algorithm exhibits significant oscillations in the final phase of training. In contrast, algorithms such as ResKPN maintain high stability throughout training. These supplementary results further demonstrate the efficacy and robustness of the proposed ResKPN algorithm.

## D.3 OTHER ABLATION EXPERIMENTS

In this section, we investigate whether the integration of a residual layer can enhance the performance or reduce the training variance of PPO and DPO algorithms. The episodic rewards during training across multiple MuJoCo environments are illustrated in Figure 15.

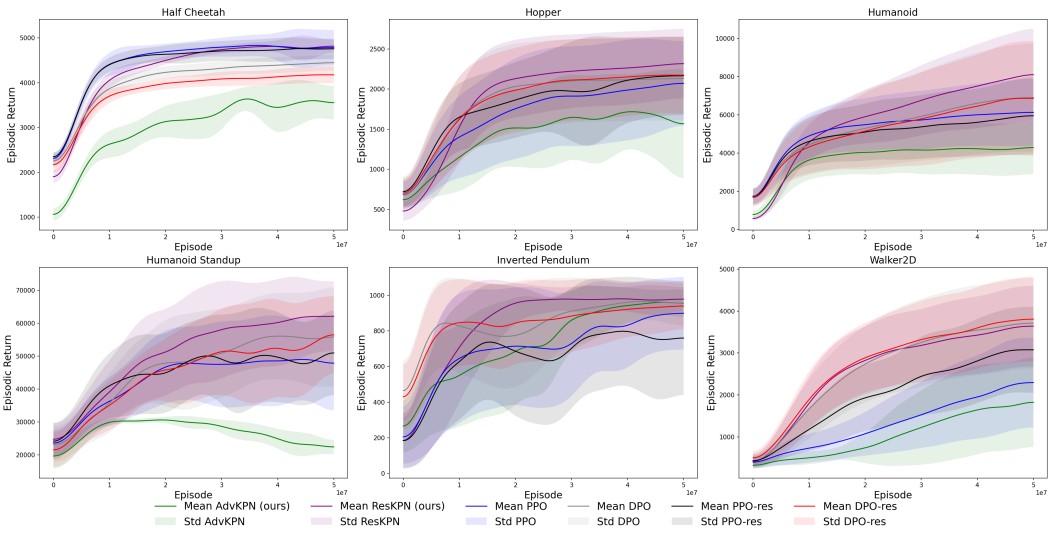

Figure 15: The episodic reward in multiple MuJoCo environments for additional baseline algorithms.

The results show that the PPO-res algorithm exhibits improvements in Walker2D and Hopper, maintains similar performance in Half Cheetah and Humanoid Standup, but experiences a decline in performance in Inverted Pendulum and Humanoid. For DPO-res, its performance remains largely

unchanged, with a slight decrease observed in Half Cheetah. A plausible explanation for this trend is that the existing neural network structures in PPO and DPO may already provide sufficiently well-learned representations, rendering the addition of a residual layer less impactful. The residual layer might only offer benefits in specific cases, such as addressing issues of overfitting or gradient dispersion. However, it does not function as an additional representation learning component, as it does in ResKPN, to further enhance performance.

The variance reduction effect of integrating the residual layer with AdvKPN is another significant consideration. The standard deviations during training for the additional baseline algorithms are presented in Figure 16.

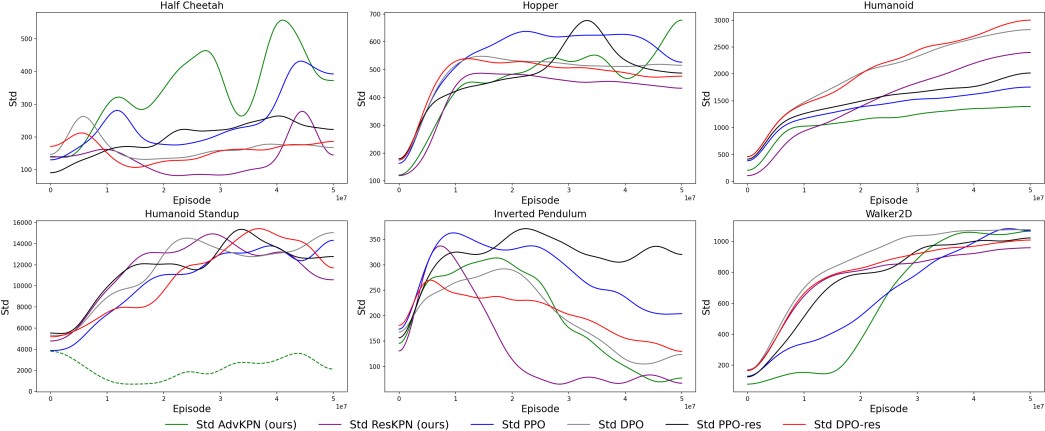

Figure 16: The standard deviation in multiple MuJoCo environments for additional baseline algorithms.

As shown in Figure 16, the addition of the residual layer has minimal impact on training variance across all tested environments. Both PPO-res and DPO-res maintain similar variance levels to their respective baselines, PPO and DPO. This can be attributed to the similar gradient calculation mechanisms used in PPO, DPO, and their residual-layer variants. Adding a residual layer directly into the neural network architecture may not effectively reduce variance in these cases.

In contrast, for policies with inherently high variance, such as KPN and AdvKPN, integrating a stable learner, like the residual network, significantly stabilizes training, as demonstrated in our theoretical analysis and the visualizations provided in Appendix C. This highlights the effectiveness of the residual layer in scenarios where the original policy struggles with variance-related instability.

## D.4 ALTERNATIVE KERNEL CHOICES

In this section, we evaluate alternative kernel choices within the ResKPN algorithm by testing Linear, Laplacian, and Sigmoid kernels, and comparing their performance against the Gaussian kernel and the PPO algorithm. The episodic rewards achieved by ResKPN with different kernels across multiple MuJoCo environments are presented in Figure 17. As shown in the figure, the Gaussian kernel achieves the best overall performance, with the Laplacian kernel yielding comparable episodic rewards. The Sigmoid kernel performs slightly worse but still outperforms PPO in most environments, except for Half Cheetah. The Linear kernel, however, produces the lowest performance due to its inability to capture non-linear relationships, which are critical for complex tasks requiring intricate patterns and interactions. We also examine the variance in episodic rewards, as shown in Figure 18.

Observing the figure, it is evident that ResKPN with Gaussian, Sigmoid, and Laplacian kernels exhibits similar stability, with relatively low variance across episodes in most environments. In contrast, the Linear kernel demonstrates significantly higher variance, especially in complex environments such as Half Cheetah and Hopper. This elevated variance underscores the limitations of the Linear kernel, as its inability to model non-linear relationships results in less stable and reliable performance.

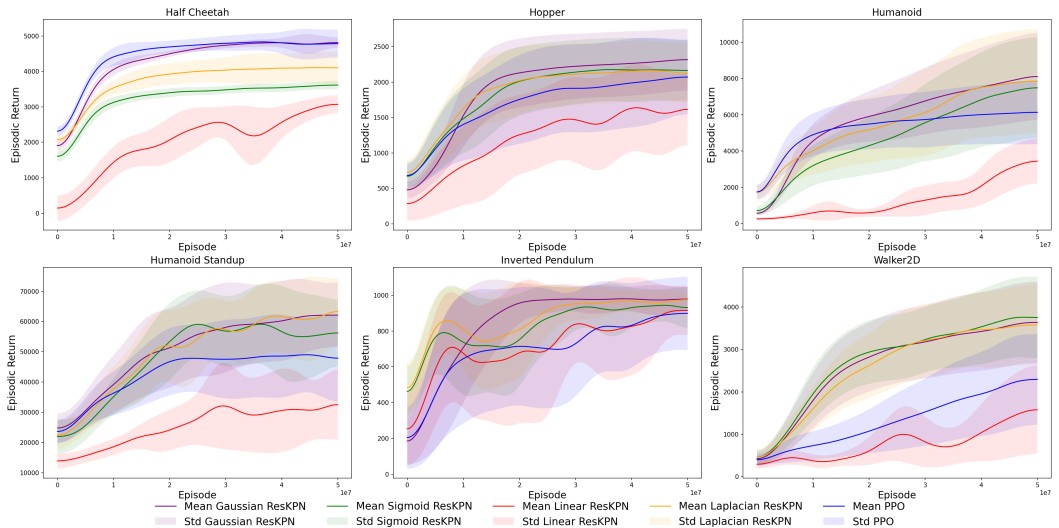

Figure 17: The episodic reward in multiple MuJoCo environments for ResKPN within different kernels.

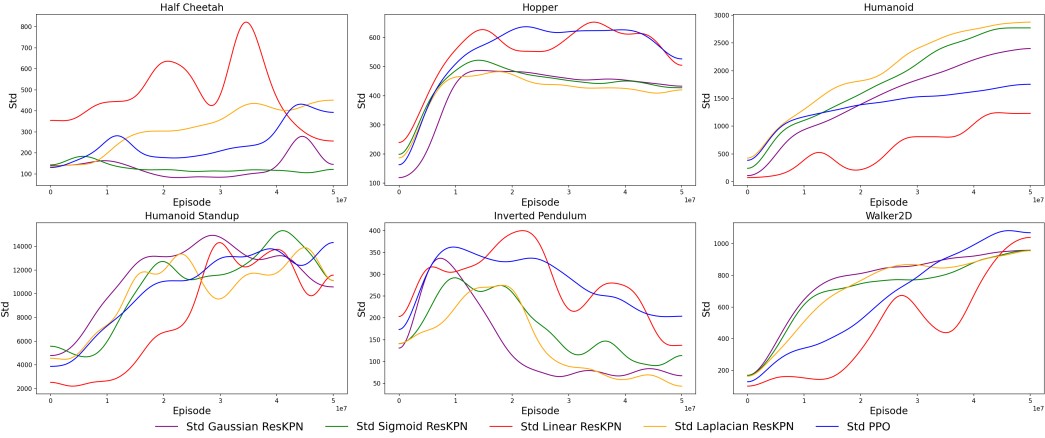

Figure 18: The standard deviation in multiple MuJoCo environments for ResKPN within different kernels.

# E   THE COMPUTATIONAL COST

In this section, we show the computational cost in Table 4. All algorithms are accelerated using JAX, significantly reducing computation time (Freeman et al., 2021). Among the tested methods, PPO and DPO exhibit the lowest overall computational costs. Integrating the RKHS policy with representation learning increases computation time due to the additional computational requirements of the RKHS function, which vary depending on the chosen kernel. Specifically, Linear ResKPN and Sigmoid ResKPN achieve moderate computation times, whereas Gaussian ResKPN and Laplacian ResKPN exhibit the highest computational costs, attributed to the intensive calculations required for these kernels.

A comparison of episodic reward performance and computational costs reveals a trade-off between performance and efficiency. For applications prioritizing policy performance, the Gaussian or Laplacian kernel may be preferred due to their superior episodic rewards. Conversely, for scenarios emphasizing computational efficiency while maintaining reasonable policy performance, the Sigmoid kernel offers a balanced alternative. The choice of kernel should be guided by the specific requirements and constraints of the application environment.

Table 4: Runtime Comparison Across Different Algorithms and Environments (in minutes).

| Environment | KPN | Gaussian ResKPN | PPO | DPO | Laplacian ResKPN | Linear ResKPN | Sigmoid ResKPN |
|---|---|---|---|---|---|---|---|
| Half Cheetah | 13.09 | 13.18 | 3.95 | 4.68 | 14.21 | 6.08 | 6.24 |
| Humanoid Standup | 13.70 | 13.77 | 2.08 | 2.14 | 12.56 | 7.37 | 6.61 |
| Inverted Pendulum | 10.29 | 10.31 | 3.50 | 3.86 | 11.22 | 2.60 | 2.63 |
| Walker2d | 11.00 | 11.06 | 3.69 | 4.12 | 10.59 | 3.55 | 3.69 |
| Hopper | 10.69 | 10.80 | 1.54 | 1.55 | 13.56 | 3.30 | 3.34 |
| Humanoid | 13.07 | 13.23 | 2.13 | 2.28 | 15.43 | 5.96 | 6.12 |

