# OpenReview forum: "Residual Kernel Policy Network: Enhancing Stability and Robustness in RKHS-Based Reinforcement Learning"
_ICLR.cc/2025/Conference — ICLR 2025 Poster_

### Official Review · Reviewer_KaJy · 2024-10-16

**Soundness:** 2
**Presentation:** 2
**Contribution:** 3
**Rating:** 3
**Confidence:** 4

**Summary:**

This paper addresses the challenges of achieving optimal performance in RL using policies modeled in reproducing RKHS. While RKHS-based methods offer efficient exploration of local optima, they suffer from significant instability due to high variance in policy gradients and sensitivity to hyperparameters. The authors analyze the causes of instability, particularly highlighting the increased gradient variance with wide-bandwidth kernels. To resolve these issues, they propose the ResKPN, a novel approach that integrates representation learning to process complex observations and introduces a residual layer inspired by advantage functions. This residual layer reduces gradient variance, thereby improving training stability and policy robustness. The ResKPN algorithm achieves state-of-the-art performance, with a 30% increase in episodic rewards across multiple complex environments.

**Strengths:**

The paper presents a clear and well-defined contribution by addressing the instability and sensitivity issues in RKHS-based reinforcement learning methods. The introduction of the ResKPN and the integration of representation learning and a residual layer provide a novel solution to these challenges. The contribution is clearly articulated, with a strong emphasis on how the proposed method improves stability and performance in complex environments. The significant 30% improvement in episodic rewards further highlights the effectiveness of the approach.

**Weaknesses:**

A notable weakness of the paper is the absence of available code for the proposed ResKPN. The lack of code limits reproducibility and hinders other researchers from validating the results or building upon the work. Providing access to the implementation would significantly enhance the paper's impact and facilitate further exploration of the proposed methods.

**Questions:**

N/A

---

> ### Author Response · Authors · 2024-11-18
> **Response to Reviewer KaJy**
>
> We acknowledge the concern regarding the absence of available code for the proposed ResKPN. To address this, we have now open-sourced the complete implementation of ResKPN along with its relevant baseline algorithms, which are provided in the supplementary material. The supplementary includes a detailed README that offers step-by-step instructions to guide you through setting up and running the project.
>
> We believe that making our code publicly available will enhance the reproducibility of our research and allow other researchers to build upon our work effectively. If you encounter any issues while reproducing our code, please feel free to reach out!

---

> > ### Comment · Reviewer_KaJy · 2024-11-26
> > **No change.**
> >
> > Same decision.

---

### Official Review · Reviewer_CJFW · 2024-10-23

**Soundness:** 3
**Presentation:** 3
**Contribution:** 3
**Rating:** 8
**Confidence:** 3

**Summary:**

This paper sets out to make a new SOTA in RL policy gradient algorithms, by modifying a method from reproducing kernel Hilbert space (RKHS) reinforcement learning, where policies are represented as Gaussians in a RKHS.  This allows policies to be learned in a space that captures relationships and correlations in large-dimensional action spaces.

The paper argues that previous RKHS RL approaches have suffered for two reasons.  First, the selection of the kernel is important and difficult, and an improper kernel choice leads to underperformance.  Second, RKHS RL is particularly vulnerable to high variance in the gradients, leading to unstable learning.

The paper addresses these issues by introducing ResKPN. This policy algorithm addresses the representation problem by applying the kernel to a learned representation of the state rather than to the state itself, and the high variance problem by introducing a residual layer in the representation to empirically decrease this variance.

The paper shows that policies learned via ResKPN outperform or compete closely with benchmark algorithms like PPO on a variety of standard RL problems.

**Strengths:**

- The problem is clearly explained.  It is clear what problems with RKHS RL the authors are setting out to fix, and how those problems motivate the produced algorithm.
- The mathematics and notation are professionally done, and are easy enough to follow (though I didn't go through the derivations in the appendices).
- The writing is clear.
- The experiments in the experimental section are comprehensive and convincing.
- A more effective RL baseline is significant... if it's usable (see weaknesses/questions).

**Weaknesses:**

- An important argument of the paper is that representation and variance problems cause RKHS RL to fail.  Accordingly, something like the illustrations of the representation and variance problems (Figure 1) are probably necessary, but I do not find these particular illustrations very effective.  They show that on one problem, some Gaussian kernels are ineffective, and that on another (single) problem, high variance can be seen.  I don't think they reinforce the strong causal relationship that the authors intend to convey, particularly when the high variance is itself dependent on the kernel selection.  Representation problems are certainly easy enough to believe, but the fact that the fully connected layer is effective *because it diminishes variance*, rather than (for example), just because it augments the representation, is not so clearly argued.
- How slow is this? Seems like it might be very slow... Is it slow enough to be near-unusable?  I think this should be addressed with a table of training times in the appendix.

**Minor things**
- The system being trained in this algorithm is complex, with lots of different sets of parameters ($\theta, \iota, \delta...$).  I think Figure 6 is important enough that it should probably be promoted to the regular paper, as the explanation is not clear enough to stand on its own without it.  The critic network should also be integrated into this figure.
- On line 96, $U(w)$ should be $U(\pi_w)$.
- On line 191, should be $\sigma^2=0.3, 0.5, 0.7,$ and 0.9.
- Wording on lines 262-263 is not correct.
- On line 299, "The key idea of residual layer" should be "They key idea of the residual layer" (or "motivating the residual").

**Questions:**

- How slow is this?  Please provide some training time comparisons with PPO.
- Do you have any further explanation or intuition for the variance-kills-RKHS-methods argument?  Would minibatches mitigate this?

---

> ### Author Response · Authors · 2024-11-18
> **Response to Reviewer CJFW (1/3)**
>
> We greatly value your thoughtful feedback on our research and have carefully addressed your comments. The corresponding revisions are highlighted in blue for clarity. Thank you for your detailed review and for supporting our work. Below, we provide responses to your specific concerns and questions.
>
> ---
>
> ## For Weaknesses
>
> > **An important argument of the paper is that representation and variance problems cause RKHS RL to fail. Accordingly, something like the illustrations of the representation and variance problems (Figure 1) are probably necessary, but I do not find these particular illustrations very effective. They show that on one problem, some Gaussian kernels are ineffective, and that on another (single) problem, high variance can be seen. I don't think they reinforce the strong causal relationship that the authors intend to convey, particularly when the high variance is itself dependent on the kernel selection.**
>
> response: Thank you for your thoughtful feedback on Figure 1. To improve its clarity and persuasiveness, We now test both subplots (a) and (b) in the Inverted Pendulum environment to ensure consistency and enhance the figure's interpretability. Specifically, we observe that when $\sigma^2 = 0.01$ and $5.0$, the RKHS policy fails to update. In contrast, for $\sigma^2 = 0.05, 0.1, 0.5,$ and $1.0$, the RKHS policy demonstrates significant improvement in performance. We believe this consistent evaluation within the same environment strengthens the argument and makes the presented issue more compelling.
>
> We hope that these revisions better convey the causal relationship between representation and variance issues in RKHS RL. If you still find the current presentation unclear or lacking, we would greatly appreciate any further suggestions to improve it. Please feel free to share your feedback with us!
>
> > **Representation problems are certainly easy enough to believe, but the fact that the fully connected layer is effective because it diminishes variance, rather than (for example), just because it augments the representation, is not so clearly argued.**
>
> response: Thank you for highlighting this important aspect of our work. We acknowledge that our original explanation did not sufficiently clarify how the fully connected layer contributes not only to variance reduction but also to enhancing representation learning. While we emphasize that the fully connected layer effectively reduces the variance of the RKHS gradient, we have revised the manuscript to better articulate its dual role (as described in Theorem 3.2 on Page 7).
>
> To make the mechanism by which the residual neural network reduces variance in RKHS policies more intuitive, we have included additional explanations following Theorem 3.2 and visual illustrations in Appendix C. These revisions provide a clearer understanding of the fully connected layer’s contribution to the variance reduction process.
>
> The design of the fully connected layer is conceptually inspired by the advantage function, which uses a stable state value function $V(\cdot)$ to stabilize the Q-function. Similarly, integrating the stable residual layer $\mu_\iota(\cdot)$ with the RKHS function $h(\cdot)$ achieves a stabilizing effect on the RKHS gradient, as supported by both theoretical analysis and visual demonstrations. Furthermore, the enhancement of representation learning is explicitly demonstrated in the experiments.
>
> We hope these revisions clarify the dual contributions of the fully connected layer and address your concerns more comprehensively. Should you have further suggestions, we would greatly appreciate your feedback.

---

> ### Author Response · Authors · 2024-11-18
> **Response to Reviewer CJFW (2/3)**
>
> > **How slow is this? Seems like it might be very slow... Is it slow enough to be near-unusable? I think this should be addressed with a table of training times in the appendix.**
>
> response: Thank you for your thoughtful comment regarding the computational cost and potential usability challenges of our method. We acknowledge that RKHS-based policies introduce additional computational overhead, particularly when compared to traditional reinforcement learning algorithms. To address this, we have added a table in Appendix E detailing the training times for our method across different kernels and environments, along with the corresponding training performances provided in Appendix D.4. These additions offer a comprehensive comparison of computational costs and highlight the trade-off between performance and efficiency.
>
> From our experiments, we observed that the computational cost varies significantly depending on the choice of kernel. For example, Gaussian and Laplacian kernels, while delivering the best performance in terms of episodic rewards, incur the highest computational costs. Conversely, Sigmoid kernels strike a balance by achieving competitive performance—lower than Gaussian ResKPN but higher than PPO—while substantially reducing training times, making them more practical for scenarios where computational efficiency is prioritized. Linear kernels, though computationally efficient, demonstrate poorer performance due to their inability to capture non-linear relationships effectively.
>
> We believe that the usability of RKHS-based policies, particularly in computationally intensive scenarios, depends on the application’s specific requirements. In settings where performance is critical and computational resources are sufficient, Gaussian or Laplacian kernels may be the preferred choice. For applications where efficiency is paramount, Sigmoid kernels provide a viable alternative.
>
> Furthermore, as noted in our conclusion, kernel-based methods remain an active area of research in machine learning and statistical learning. Significant efforts are being made to reduce their computational complexity while preserving their non-parametric modeling advantages. To this end, we plan to explore kernel embedding techniques and efficient kernel approximations as part of our future work.
>
> ## For Minor Things
>
> > **The system being trained in this algorithm is complex, with lots of different sets of parameters $(\theta, \iota, \delta \ldots)$. I think Figure 6 is important enough that it should probably be promoted to the regular paper, as the explanation is not clear enough to stand on its own without it. The critic network should also be integrated into this figure.**
>
> response: Thank you for your valuable suggestion regarding Figure 6 and its placement in the paper. Based on your feedback, we have moved Figure 6 (now Figure 2) into the main body of the paper to ensure that it serves as a clear reference for understanding the algorithm's complexity and the relationships among its various components, including the parameter sets $(\theta, \iota, \delta, \ldots)$.
>
> Additionally, we have updated the figure to better integrate the critic network into the overall scheme. The updated figure now clearly illustrates the computation of the advantage value $A^{\pi_\varpi}(a_k, s_k)$, which combines the state value $V(s_k)$ from the critic network with the reward $r(a_k, s_k)$. This integration aims to provide a more comprehensive and intuitive understanding of the algorithm's structure and parameter interactions.
>
> To maintain consistency and provide additional technical insights, we have also retained this figure in Appendix B.1. This ensures that readers who wish to explore the detailed methodological aspects and supplementary information can easily reference it there.
>
> > **For "On line 191 , should be $\sigma^2=0.3,0.5,0.7$, and $0.9$" .**
>
> response: We have updated Figure 1a to align with the environment used in Figure 1b. Consequently, the training performance when varying $\sigma$ has changed due to the updated environment. We have also revised the corresponding description in the manuscript to reflect these changes accurately.
>
> > **For other minor things.**
>
> response: We sincerely thank you for your meticulous review of our manuscript. We have carefully addressed and corrected the issue you pointed out. Once again, we deeply appreciate the time and effort you dedicated to providing this valuable feedback.
>
> ## For Questions
>
> > **How slow is this? Please provide some training time comparisons with PPO.**
>
> response: Thank you for your follow-up question. We have included a detailed comparison of training times between ResKPN across different kernels and PPO in Appendix E. This highlights a trade-off between computational cost and performance, with Sigmoid kernels providing a practical middle ground for efficiency and effectiveness.

---

> ### Author Response · Authors · 2024-11-18
> **Response to Reviewer CJFW (3/3)**
>
> > **Do you have any further explanation or intuition for the variance-kills-RKHS-methods argument? Would minibatches mitigate this?**
>
> response: Thank you for your insightful question regarding the variance-kills-RKHS-method argument and the potential role of minibatches in mitigating this issue. We provide two intuitive explanations for this phenomenon:
>
> (a) In our paper, the RKHS policy is updated using the stochastic gradient approach, where data samples are randomly selected from the buffer for each update. Stochastic gradients are inherently associated with higher variance compared to minibatch or batch gradients, as extensively demonstrated in [1]. Minibatch gradients reduce variance by averaging the gradients of multiple samples, leading to more stable updates. This inherent instability of stochastic gradients contributes significantly to the challenges observed in the training of RKHS policies.
>
> (b) Beyond the natural instability of stochastic gradients, as demonstrated in Lemma 3.1 and Figure 1 of our paper, RKHS gradients inherently exhibit higher variance compared to linear policies. This amplifies the instability in the stochastic RKHS gradient updates, further exacerbating the challenges in training.
>
> The impact of variance on RKHS methods primarily stems from the extreme instability of stochastic RKHS gradients, which is even more pronounced than the instability observed in Euclidean-space stochastic gradients. We appreciate your suggestion regarding minibatches, as they are widely used in neural networks and other machine learning methods to address the shortcomings of stochastic gradients by averaging gradients within the minibatch, thereby reducing variance and stabilizing training.
>
> However, applying minibatches directly to RKHS gradients poses significant computational challenges. The mean of RKHS gradients must be explicitly represented as $\sum_{i=1}^n \alpha_i K(s_i, \cdot)$, where $n$ is the minibatch size, leading to a quadratic increase in computational complexity with the minibatch size. This makes minibatch gradients computationally expensive for RKHS methods. While some approaches have been proposed to address this, such as the model-based method in [2], which averages gradients of symmetric transitions with a minibatch size of $n=2$, and the kernel matching pursuit method in [3], which reduces gradient variance through gradient regression, both approaches face limitations in complex environments like MuJoCo. Specifically, [2] struggles to identify symmetric transitions, and [3] exhibits significant computational instability in high-dimensional tasks.
>
> In our work, we opted for stochastic RKHS gradients to avoid the substantial computational complexities associated with minibatch gradients. Notably, we observed that introducing the residual network significantly reduces the variance of RKHS gradients without imposing additional computational overhead. This provides a scalable and effective solution for learning in complex environments.
>
> Following your suggestion, we included experiments testing minibatch gradients in Appendix D.1 to further investigate this approach, alongside a discussion of variance reduction techniques introduced in prior research. To evaluate the impact of minibatch size on computational cost, we conducted experiments with varying minibatch sizes in two environments, as summarized in Table 1.
> #### Table 1: The conputation time for varying minibatch sized in Half Cheetah and Hopper
> | **Environment** | **n = 1** | **n = 2** | **n = 3** | **n = 4** | **n = 5** |
> |------------------|-----------|-----------|-----------|-----------|-----------|
> | Half Cheetah     | 13.19     | 61.25     | 140.74    | 199.49    | 205.07    |
> | Hopper           | 10.83     | 58.91     | 123.12    | 169.03    | 202.61    |
>
> The results demonstrate that training time increases quadratically with minibatch size in both the Hopper and Half Cheetah environments. For instance, increasing the minibatch size to $n=5$ required over three additional hours of computation compared to $n=1$, highlighting the computational infeasibility of minibatch gradients in RKHS-based methods for high-dimensional environments.
>
> Despite this, minibatches remain crucial due to their ability to reduce training variance. To explore this, we conducted experiments using a minibatch size of $n=2$, as larger sizes were computationally prohibitive. The results in Appendix D.1 reveal that using $n=2$ minibatches effectively reduces the variance of both AdvKPN and ResKPN. This underscores the potential of minibatches in stabilizing RKHS policy methods.
>
> If computational cost constraints can be addressed, we believe that leveraging minibatches offers a promising direction for further reducing variance in RKHS-based reinforcement learning methods. This remains an important avenue for future research.
>
> We hope this response addresses your concerns, and we are happy to provide additional clarification if needed.

---

> > ### Author Response · Authors · 2024-11-18
> > **References for our response**
> >
> > [1] Qian, X., & Klabjan, D. (2020). The impact of the mini-batch size on the variance of gradients in stochastic gradient descent. arXiv preprint arXiv:2004.13146.
> >
> > [2] Paternain, S., Bazerque, J. A., Small, A., & Ribeiro, A. (2020). Stochastic policy gradient ascent in reproducing kernel hilbert spaces. IEEE Transactions on Automatic Control.
> >
> > [3] Le, T. P., Ngo, V. A., Jaramillo, P. M., & Chung, T. (2019). Importance sampling policy gradient algorithms in reproducing kernel hilbert space. Artificial Intelligence Review.

---

> > > ### Comment · Reviewer_CJFW · 2024-11-26
> > > **Response to the response...**
> > >
> > > Thank you, the changes helped with the explanations considerably.  The computational cost is bad, but not quite as bad as I feared.  I enjoyed the paper.
> > >
> > > Some very minor things:
> > > - On lines 323-324, the new text is not worded correctly.
> > > - On line 349, "effectively minimized" should be "decreased."

---

> > > > ### Author Response · Authors · 2024-11-27
> > > > **Response to Reviewer CJFW**
> > > >
> > > > Thank you once again for your thoughtful feedback and for taking the time to carefully review our paper. We are truly honored to have your support and encouragement.
> > > >
> > > > We have addressed the two very minor issues you pointed out:
> > > >
> > > > The text on lines 323–324 has been revised for clarity.
> > > >
> > > > On line 349, “effectively minimized” has been corrected to “decreased.”
> > > >
> > > > We deeply appreciate your insights and the constructive suggestions that have significantly improved the quality of our work.

---

### Official Review · Reviewer_sr1j · 2024-11-02

**Soundness:** 3
**Presentation:** 3
**Contribution:** 3
**Rating:** 8
**Confidence:** 4

**Summary:**

The paper, titled "Residual Kernel Policy Network: Enhancing Stability and Robustness in RKHS-Based Reinforcement Learning," addresses the instability and sensitivity in RKHS-based reinforcement learning policies. The authors show significant gradient variance and hyperparameter sensitivity and propose the Residual Kernel Policy Network (ResKPN). This network incorporates representation learning to adaptively align observations with the kernel's structure. The Authors also employ a residual architecture to further stabilize training. Experiments on MuJoCo tasks demonstrate ResKPN's performance, reportedly surpassing baseline algorithms like PPO and DPO by up to 30% in episodic rewards.

**Strengths:**

The technical claims are well-founded, and the experimental results are robustly supported by rigorous methodology. The integration of residual layers with RKHS gradients appears to reduce gradient variance, as confirmed by extensive empirical evidence on MuJoCo environments. The variance analysis is theoretically grounded, and experimental setups align well with the claims, ensuring soundness across technical aspects.

The presentation is clear overall, though there are instances where dense technical language or unclear phrasing makes comprehension difficult, especially in theoretical sections. Improved structuring or additional context around complex derivations could enhance readability.

This work contributes meaningfully to reinforcement learning research by empirically identifying a weakness in a common reinforcement learning approach. It attempts to solve this by introducing a model with enhanced stability and robustness through representation learning and a residual layer. The originality lies in effectively merging RKHS gradient variance reduction with neural network-based feature extraction, a strategy not previously well-addressed. The approach is promising for applications requiring adaptive, high-dimensional policy learning. However, just adding a residual neural network to an existing method has limited originality.

- Significance: Tackling gradient variance in RKHS-based reinforcement learning is critical for real-world applications, and the results demonstrate potential for improved robustness.
- Experimental Rigor: Extensive tests across six MuJoCo tasks validate ResKPN’s efficacy and its edge over comparable baselines in terms of episodic rewards and convergence rates.
- Practical Impact: The adaptability of ResKPN to complex, high-dimensional environments shows promise for real-world reinforcement learning scenarios.

**Weaknesses:**

Complexity of Variance Analysis: While theoretically thorough, the variance analysis may benefit from simplification or additional visual explanations. This complexity could present a barrier for researchers less familiar with RKHS.

Computational Cost: Given the use of RKHS, the method may face scalability limitations in more extensive settings or when applied to multi-agent environments.

Limited Discussion on Alternative Kernels: While Gaussian kernels are utilized effectively, the paper could explore the feasibility of other kernels or adaptive kernel selection strategies to further broaden the model's applicability.

**Questions:**

Could the authors expand on how ResKPN might handle multi-agent or cooperative environments? Given the scalability challenges, it would be valuable to understand the model's limitations in such settings. How would the approach adapt to environments where action spaces vary significantly in scale or complexity? Neural networks often succeed in settings with large amount of training data, would such a setting be appropriate for a non-parametric method such like RKHS? 106: h is a functional (function -> values), but notation h(s) is used, s is a state, not a function so why do we call h a functional?

---

> ### Author Response · Authors · 2024-11-18
> **Response to Reviewer sr1j (1/3)**
>
> We sincerely appreciate your insightful feedback on our work. We have made revisions accordingly, which are highlighted in blue for your convenience. Thank you for your support of our research! We present our response to each of your concerns and questions below.
>
> ---
>
> > **The approach is promising for applications requiring adaptive, high-dimensional policy learning. However, just adding a residual neural network to an existing method has limited originality.**
>
> We agree that simply adding a residual neural network to a policy modeled by a neural network may have limited originality and would not significantly impact overall variance reduction or performance enhancement. However, in our work, the policy is modeled as a high-variance RKHS policy, where the addition of a residual neural network becomes essential. This significance arises because, if the output of the residual network is more stable than that of the RKHS function $h(\cdot)$ (as demonstrated in our proofs), the variance of the RKHS policy can be reduced. Therefore, we argue that introducing a residual neural network has a novel impact specifically within the context of high-variance RKHS policies.
>
> ## For Weaknesses
>
> > **Complexity of Variance Analysis: While theoretically thorough, the variance analysis may benefit from simplification or additional visual explanations. This complexity could present a barrier for researchers less familiar with RKHS.**
>
> response: Thank you for your valuable feedback regarding the complexity of the variance analysis. To address this, we have refined the derivations in the variance analysis section by introducing the notations $\Gamma_A$, $\Gamma_V$, and $\Gamma_{h,\mu}$, which significantly simplify the mathematical formulation and enhance the overall readability. Additionally, to make the mechanism by which the residual neural network reduces the variance of RKHS policies more accessible (beyond the explanation in Theorem 3.2), we have included more intuitive interpretations following the theorem and added visual explanations in Appendix C. Conceptually, the introduction of the residual network is inspired by the advantage function's use of a stable state value function $V(\cdot)$ to stabilize the Q-function. Similarly, by combining the stable residual layer $\mu_\iota(\cdot)$ with the RKHS function $h(\cdot)$, we aim to achieve a stabilizing effect on the RKHS gradient, as demonstrated both theoretically and visually.
>
> > **Computational Cost: Given the use of RKHS, the method may face scalability limitations in more extensive settings or when applied to multi-agent environments.**
>
> response: Thank you for your insightful comment regarding the computational cost and scalability challenges of RKHS policies in more extensive settings or multi-agent environments. We acknowledge that the integration of RKHS policies introduces additional computational overhead, which may pose limitations in such scenarios. We also add a table in Appendix E to show the computational cost. Kernel-based methods remain an active area of research, particularly in machine learning and statistical learning, where significant efforts have been made to reduce their computational complexity while retaining the advantages of non-parametric modeling. In this case, we believe that further optimization of RKHS-based policies can achieve a better balance between computational efficiency and improved performance over traditional reinforcement learning algorithms. To address this, we have added these potential advancements in the Conclusion section, emphasizing our future focus on exploring kernel embedding techniques and efficient kernel approximations to mitigate computational challenges and enhance scalability.

---

> ### Author Response · Authors · 2024-11-18
> **Response to Reviewer sr1j (2/3)**
>
> > **Limited Discussion on Alternative Kernels: While Gaussian kernels are utilized effectively, the paper could explore the feasibility of other kernels or adaptive kernel selection strategies to further broaden the model's applicability.**
>
> response: Thank you for your valuable feedback regarding the exploration of alternative kernels. In response to your suggestion, we have conducted additional experiments incorporating Laplacian, Sigmoid, and Linear kernels as alternative choices. The corresponding results have been included in Appendix D.4 and Appendix E. By comparing training performance and computational cost, we observed a trade-off between performance and efficiency across different kernels.
>
> Specifically, Gaussian and Laplacian kernels achieve the best training performance but incur the highest computational costs. Sigmoid kernels demonstrate moderate training performance (outperforming PPO in most environments) while significantly reducing computational requirements. In contrast, Linear kernels perform the worst due to their limited ability to capture non-linear relationships.
>
> We believe that the choice of kernel should be guided by the specific requirements and constraints of the application environment. This comparison of different kernels aims to provide a more comprehensive perspective on their applicability and expand the potential application space of the model.
>
> ## For Questions
>
> > **Could the authors expand on how ResKPN might handle multi-agent or cooperative environments? Given the scalability challenges, it would be valuable to understand the model's limitations in such settings. How would the approach adapt to environments where action spaces vary significantly in scale or complexity?**
>
> response: Thank you for your insightful questions.
> **Regarding the application of ResKPN in multi-agent or cooperative environments,** we address this from two perspectives:
>
> (a) The current ResKPN framework can be seamlessly integrated into existing multi-agent algorithms, such as IPPO, MAPPO, MAAC [1], and COMA [2]. Specifically, the Actor in these algorithms can be replaced with an RKHS policy, leveraging corresponding networks for representation learning and utilizing the Q-value learning techniques inherent to these methods. From this perspective, ResKPN can extend and adapt to tackle multi-agent or cooperative environments using existing approaches.
>
> (b) Multi-agent problems present unique challenges, such as the curse of dimensionality and the need to effectively model cooperation among agents, which differ significantly from single-agent problems. The potential of RKHS to address these challenges remains an active area of research. For instance, recent studies [3, 4, 5] have employed kernel mean embedding and mean-field theory to significantly alleviate the dimensionality explosion in multi-agent or cooperative environments. We are actively exploring this direction and aim to apply RKHS policies creatively to address the unique challenges in multi-agent settings.
>
> Additionally, we have discussed the scalability challenges and limitations of RKHS policies in Appendix E, particularly regarding their increased computational cost. This remains one of the current limitations of our work. (The following response for this question is in the next comment due to the limitation of characters......)

---

> ### Author Response · Authors · 2024-11-18
> **Response to Reviewer sr1j (3/3)**
>
> **Regarding environments where action spaces vary significantly in scale or complexity,**
> we provide the following considerations:
>
> (a) From a computational complexity perspective, our method exhibits linear growth in computation with respect to the dimensionality of the action space. For an $n$-dimensional action space, the computational complexity increases $n$-fold compared to a single-dimensional action space. This is due to the RKHS gradient formulation:
>
> $$
> \nabla_h \hat{U}\left(\pi_h\right) = \eta K(s_k, \cdot) \boldsymbol{\Sigma}^{-1}(a_k - h(s_k)) \hat{Q}^{\pi_h}(a_k, s_k),
> $$
>
> where $a_k$ being an $n$-dimensional vector results in the RKHS gradient also being $n$-dimensional. This linear scalability explains the strong performance of our method in high-dimensional single-agent environments, such as Humanoid (17-dimensional action space). As demonstrated in Appendix E, the computational time remains largely unaffected by increasing action space dimensions. However, for environments with even higher-dimensional action spaces, our method may still face challenges.
>
> (b) From a complexity modeling perspective, high-dimensional action spaces often exhibit strong correlations among different action dimensions. Using the current RKHS gradient formulation directly may fail to fully capture these correlations. Existing literature suggests the use of a modified gradient:
>
> $$
> \nabla_h \hat{U}\left(\pi_h\right) = \eta K(s_k, \cdot) \boldsymbol{\Sigma}^{-1} A (a_k - h(s_k)) \hat{Q}^{\pi_h}(a_k, s_k),
> $$
>
> where $A$ is an $n \times n$ matrix that can be either learnable or predefined based on prior knowledge. However, in our preliminary tests, this approach did not significantly improve overall performance. Effectively modeling the complexity of correlations within action dimensions in RKHS policies remains an open question and a key direction for our future research.
>
> We hope these discussions address your concerns and provide clarity regarding the potential applications and limitations of ResKPN in multi-agent and high-dimensional action space environments.
>
> > **Neural networks often succeed in settings with large amount of training data, would such a setting be appropriate for a non-parametric method such like RKHS?**
>
> response: Thank you for your insightful question. We believe the key difference between gradient updates in neural networks and RKHS policies in the context of large training datasets lies in the use of mini-batch gradients. Neural networks leverage mini-batch gradients to compute stable and efficient updates, enabling a smooth gradient descent process even with large datasets. In contrast, for RKHS policies—particularly those in continuous spaces—each data point corresponds to an independent kernel function. As a result, performing gradient updates with large amounts of data directly can become computationally inefficient.
>
> In our method, we adopt a stochastic gradient approach for RKHS policies, where a randomly sampled data point is used for each gradient update. Meanwhile, representation learning within our framework utilizes mini-batch gradients to maintain computational efficiency. To address the challenge of efficiently utilizing large datasets in RKHS, some studies have proposed methods such as recursive least squares [6] to approximate RKHS gradients, achieving promising results.
>
> We believe there is potential for further advancements in this area to balance the computational efficiency of RKHS with its sample efficiency. We have acknowledged this in our discussion of future work, where we aim to explore more effective solutions for scaling RKHS-based methods in settings with large training datasets.
>
> > **106: h is a functional (function $->$ values), but notation h(s) is used, s is a state, not a function so why do we call h a functional?**
>
> response: Thank you for pointing out this important detail. You are correct that $h$ is a function, and our description in the manuscript was inaccurate. We appreciate your careful observation and have revised the text to correctly reflect that $h$ is a function, not a functional.  We apologize for any confusion caused and thank you for bringing this to our attention.

---

> ### Author Response · Authors · 2024-11-18
> **References for our response**
>
> [1] Du, Y., Leibo, J. Z., Islam, U., Willis, R., & Sunehag, P. (2023). A review of cooperation in multi-agent learning. arXiv preprint arXiv:2312.05162.
>
> [2] Foerster, J., Farquhar, G., Afouras, T., Nardelli, N., & Whiteson, S. (2018). Counterfactual multi-agent policy gradients. In Proceedings of the AAAI conference on artificial intelligence.
>
> [3] Chen, M., Li, Y., Wang, E., Yang, Z., Wang, Z., & Zhao, T. (2021). Pessimism meets invariance: Provably efficient offline mean-field multi-agent RL. Advances in Neural Information Processing Systems.
>
> [4] Liu, J., & Lian, H. (2024). Kernel-Based Decentralized Policy Evaluation for Reinforcement Learning. IEEE Transactions on Neural Networks and Learning Systems.
>
> [5] Bukharin, A., Li, Y., Yu, Y., Zhang, Q., Chen, Z., Zuo, S., ... & Zhao, T. (2024). Robust multi-agent reinforcement learning via adversarial regularization: Theoretical foundation and stable algorithms. Advances in Neural Information Processing Systems.
>
> [6] Alipoor, G., & Skretting, K. (2023). Kernel recursive least squares dictionary learning algorithm. Digital Signal Processing.

---

### Official Review · Reviewer_eyTM · 2024-11-02

**Soundness:** 2
**Presentation:** 3
**Contribution:** 2
**Rating:** 5
**Confidence:** 3

**Summary:**

This paper applies Reproducing Kernel Hilbert Space (RKHS) methods to policy gradient to enhance sample efficiency and stability in training. Additionally, it introduces a variance reduction technique inspired by residual networks, further improving the stability and effectiveness of the policy training process.

**Strengths:**

1. This paper introduce a new RKHS policy learning algorithm.
2. This paper introduces a variance reduction technique by designing a residual layer for the RKHS policy
3. The numerical results demonstrate the validity of the proposed method

**Weaknesses:**

1. While applying RKHS to reinforcement learning (RL) is not novel, this paper lacks a discussion of existing methods. Relevant references include:
[1] Mazoure, Bogdan, et al. "Representation of reinforcement learning policies in reproducing kernel Hilbert spaces." arXiv preprint arXiv:2002.02863 (2020).
[2] Wang, Yiwen, and Jose C. Principe. "Reinforcement learning in reproducing kernel Hilbert spaces." IEEE Signal Processing Magazine 38.4 (2021): 34-45.
Additionally, some kernel-based methods, although not specifically RKHS-based, are also relevant to consider.
2. Existing work, such as reference [2], introduces variance reduction techniques. A comparison or discussion of these approaches with the methods in this paper would provide valuable insights. Although RKHS is rarely applied to RL, there is extensive work on integrating RKHS with general machine learning problems.
3. The idea of applying RKHS to RL appears straightforward, and the key distinctions from previous approaches remain unclear.

**Questions:**

1. Based on the numerical results, it appears that the main improvement stems from the residual design. However, the comparison models are baseline models without any variance reduction techniques, raising questions about the fairness of the comparison. Additionally, variance reduction methods introduced in previous works should be considered.
2. There is existing literature on combining RKHS with residual networks, and a discussion of these studies would add valuable context.
3. The numerical section would benefit from testing in more complex environments to strengthen the evaluation.

---

> ### Author Response · Authors · 2024-11-18
> **Response to Reviewer eyTM (1/5)**
>
> We sincerely appreciate your detailed feedback and insightful comments. To address your concerns, we have conducted a comprehensive review of related works, supplemented our experiments, and expanded our discussion to clarify the novelty and contributions of our approach. The revisions are clearly highlighted in blue in the updated manuscript. Below, we provide detailed responses to your comments.
>
> ---
>
> ## For Weaknesses
> > **1. While applying RKHS to reinforcement learning (RL) is not novel, this paper lacks a discussion of existing methods. Relevant references include: [1] Mazoure, Bogdan, et al. "Representation of reinforcement learning policies in reproducing kernel Hilbert spaces." arXiv preprint arXiv:2002.02863 (2020). [2] Wang, Yiwen, and Jose C. Principe. "Reinforcement learning in reproducing kernel Hilbert spaces." IEEE Signal Processing Magazine 38.4 (2021): 34-45. Additionally, some kernel-based methods, although not specifically RKHS-based, are also relevant to consider.**
>
> response: Thank you for providing these additional references. We have thoroughly reviewed both papers and provide a detailed comparison below:
>
> (a) Regarding [1], the paper proposes truncating RKHS embeddings to represent policies as ${\hat\pi_{K}}(a \mid s) = \sum_{k=1}^K \xi_k \omega_k(s, a)$ and theoretically demonstrates that the Q-function variance under the truncated RKHS policy ${\hat \pi_{K}}$ is smaller than that of the original policy, i.e., $\mathbb V_\beta\left[Q^\pi\right] \geq \mathbb V_\beta\left[Q^{\hat\pi_K}\right]$. While this method offers strong theoretical insights, its implementation relies on partitioning the state space into fixed bins and introducing kernel orthogonal basis for algorithmic iterations. As noted in the paper: *"We assume that state components assigned to the same bin have a similar behavior, a somewhat limiting condition which is good enough for simple environments and greatly simplifies our proposed algorithm."* Consequently, this approach is practical only in simple environments and cannot be directly applied to the complex MuJoCo testing environments in our paper. For example, in the Hopper environment with a 17-dimensional state space, even a simple binary division per dimension results in $2^{17}=131072$ kernel orthogonal basis, making the approach computationally infeasible for our use case.
>
> (b) Regarding [2], this paper applies RKHS-based RL methods to decoder design, focusing on building optimal, universal neural-action mappings with significant value in neural signal processing. The paper also proposes an online clustering method to aggregate similar kernel orthogonal bases into central representations, thereby reducing computational costs and variance. However, several characteristics of this method limit its applicability to our current algorithm:
> 1. The method primarily focuses on learning the Q-function in RKHS, while our approach is centered on policy gradient optimization in RKHS.
> 2. The clustering technique introduces bias into the RKHS policy gradient, resulting in a bias-variance trade-off. Determining how to implement this clustering method effectively within our RKHS policy gradient framework remains an open question.
>
> We have incorporated a discussion of both papers into Section 1 (Introduction) of the revised manuscript and sincerely thank you for suggesting these valuable references.
>
> Finally, regarding your point on kernel-based methods that may not specifically rely on RKHS, since each kernel can form a Reproducing Kernel Hilbert Space, we address this as follows:
>
> (a) For kernel-based methods that do not involve RKHS theory, we discuss relevant approaches in Section 2.2 under Deep Kernel Learning. Since our paper focuses on neural network integration, we have primarily compared kernel-based methods with deep learning-related work.
>
> (b) For methods using feature maps that resemble kernel-based approaches but do not form an RKHS, we note that our RKHS gradient $\nabla_h \hat U\left(\pi_h\right) = \eta K(s_k, \cdot) \boldsymbol{\Sigma}^{-1}(a_k-h(s_k)) \hat Q^{\pi_h}(a_k, s_k)$ critically relies on the reproducing property of RKHS. Without this property, the gradient cannot be derived as presented in our paper. Therefore, we believe the scope of this paper should remain focused on RKHS-based techniques.
>
> Thank you again for your constructive feedback and for bringing these references to our attention. We hope our response clarifies our approach and rationale. Please let us know if further elaboration is needed.

---

> ### Author Response · Authors · 2024-11-18
> **Response to Reviewer eyTM (2/5)**
>
> > **2. Existing work, such as reference [2], introduces variance reduction techniques. A comparison or discussion of these approaches with the methods in this paper would provide valuable insights. Although RKHS is rarely applied to RL, there is extensive work on integrating RKHS with general machine learning problems.**
>
> response: We sincerely appreciate your thoughtful suggestions and comments. In our revised manuscript, we have expanded our discussion of variance reduction methods to address the literature you mentioned, as well as additional approaches that we referenced in the Introduction section of our paper but had not previously detailed. Below, we summarize the key variance reduction methods for RKHS-based reinforcement learning policies, including the references [1] and [2], which are also presented in the following Table 1 for clarity.
>
> #### Table 1: Variance Reduction Methods for RKHS Policies
> | **Paper**         | **Variance Reduction Method**              | **Tested Environment**                   | **State** | **Action** |
> |--------------------|--------------------------------------------|------------------------------------------|-----------|------------|
> | [3]               | Symmetric estimation                      | Self-Charging Surveillance Robot         | 3         | 2          |
> | [4]               | Kernel matching pursuit                   | Quadrotor Navigation                     | 13        | 3          |
> | [1]               | Predefined kernel orthogonal basis        | Pendulum                                 | 3         | 1          |
> | [2]               | Kernel orthogonal bases clustering        | Nonhuman primate performing obstacle-avoidance task | (not mentioned)       | 1          |
>
> ### Variance Reduction Methods for RKHS Policies
> - **Symmetric estimation** [3]: This method is effective in simple environments such as surveillance robots, where symmetric transitions are identifiable. However, it becomes impractical in complex settings like MuJoCo environments, where symmetric transitions are rare and challenging to identify.
> - **Kernel matching pursuit** [4]: By reducing variance through gradient regression, this method demonstrates effectiveness in relatively simple environments but suffers from computational instability and inefficiency in high-dimensional tasks, such as those involving large state-action spaces.
> - **Predefined kernel orthogonal basis** [1]: This method represents policies using kernel orthogonal bases and theoretically reduces Q-function variance. However, it requires partitioning the state space into bins, which leads to exponential growth in computational complexity in high-dimensional environments. For instance, in the Hopper environment with a 17-dimensional state space, binary partitioning results in over 131,000 bins, making this approach computationally infeasible in complex MuJoCo environments. (Detailed in the previous response)
> - **Kernel orthogonal bases clustering** [2]: This approach aggregates similar kernel orthogonal bases to reduce variance and computational costs, offering benefits in neural signal processing tasks. However, it introduces bias and focuses primarily on Q-function learning rather than policy gradient optimization, limiting its applicability to RKHS policy frameworks. (Detailed in the previous response)
>
>
> While these methods provide valuable insights, they face significant limitations when applied to the high-dimensional, continuous control tasks commonly found in MuJoCo environments. We also included a discussion and comparison of minibatch gradient methods [5], which are widely used in machine learning. However, our experiments revealed that this approach leads to computational time growing quadratically with the minibatch size, making it challenging to apply in high-dimensional environments like those in MuJoCo. The corresponding computational time analysis and experimental results have been added to Appendix D.1.

---

> ### Author Response · Authors · 2024-11-18
> **Response to Reviewer eyTM (3/5)**
>
> > **3. The idea of applying RKHS to RL appears straightforward, and the key distinctions from previous approaches remain unclear.**
>
> response: Thank you for your valuable feedback. We fully agree that directly applying RKHS to reinforcement learning is indeed a straightforward concept, and we do not argue this as the novelty of our work. Instead, our focus lies in addressing key challenges within the realm of RKHS policy-based methods, distinguishing them from value-based approaches.
>
> To clarify, RKHS RL research spans two primary directions: value-based and policy-based methods. Value-based approaches, such as those in [2], utilize RKHS to approximate Q-functions, analogous to leveraging neural networks for non-linear representations. In contrast, policy-based methods, as explored in [1],[3],[4], model policies directly within RKHS spaces and optimize them through policy gradients, which is the focus of our paper. While policy-based RKHS methods are not inherently novel, they present unique challenges, particularly in high-dimensional, complex environments like MuJoCo.
>
> The critical challenges that remain unresolved in RKHS policy-based methods are as follows:
>
> 1. **Robustness**: RKHS policies exhibit extreme hyperparameter sensitivity [6], as demonstrated in Figure 1a. This makes it challenging to generalize across environments or tune parameters effectively in complex settings.
> 2. **Stability**: High variance in RKHS policy gradients leads to instability during training [4], as illustrated in Figure 1b, hindering convergence to optimal policies.
>
> Our work, **Residual Kernel Policy Network: Enhancing Stability and Robustness in RKHS-Based Reinforcement Learning**, aims to address these challenges through the following contributions:
>
> **For Robustness:**
> 1. We propose a novel RKHS policy learning framework that employs a neural network for dynamic observation representation, aligning the input distribution with the kernel to improve robustness across environments.
>
> **For Stability:**
>
> 2. We conduct a detailed variance analysis of RKHS policy gradients, highlighting how wide-bandwidth kernels exacerbate instability and variance in traditional RKHS policies.
> 3. We introduce a residual layer to reduce RKHS gradient variance. This innovation stabilizes training while enhancing performance, enabling ResKPN to achieve a 30\% improvement in episodic rewards in the Humanoid environment.
>
> These contributions aim to resolve the aforementioned challenges and advance RKHS policy-based reinforcement learning in complex scenarios. We hope this explanation clarifies the scope and novelty of our work.

---

> ### Author Response · Authors · 2024-11-18
> **Response to Reviewer eyTM (4/5)**
>
> ## For Questions
> > **1. Based on the numerical results, it appears that the main improvement stems from the residual design. However, the comparison models are baseline models without any variance reduction techniques, raising questions about the fairness of the comparison. Additionally, variance reduction methods introduced in previous works should be considered.**
>
> response: Thank you for your valuable feedback. Regarding your observation that the main improvement stems from the residual design, we partially agree but would like to clarify that our improvements can be understood from two perspectives:
>
> 1. **Compared to existing RKHS policy methods (e.g., Origin-Kernel):** The improvements in our method are achieved through three progressive advancements—representation learning (KPN algorithm), the incorporation of advantage functions (AdvKPN algorithm), and the design of the residual layer (ResKPN). These steps collectively address the challenges of robustness and stability in current RKHS policies, resulting in significant performance gains.
>
> 2. **Compared to existing baseline methods (e.g., PPO, DPO):** The improvements primarily come from the integration of RKHS policies with advanced representation learning techniques and the residual design, which together outperform standard baseline methods.
>
> We hope this explanation provides a clearer understanding of the novelty of our approach.
>
> Regarding the **fairness of comparisons**, we fully agree with your concern and have conducted additional experiments to address this issue, as presented in Appendix D.3. These experiments investigate whether the residual layer enhances performance or reduces training variance in PPO and DPO algorithms. The results show that:
>
> - **Performance:** The PPO-res algorithm demonstrates improvements in Walker2D and Hopper but shows limited or declining performance in environments like Inverted Pendulum and Humanoid. For DPO-res, the performance remains largely unchanged, with minor decreases in some environments. This indicates that the existing neural network structures in PPO and DPO already learn representations effectively, and the addition of a residual layer may not always lead to further improvements, except in cases of overfitting or gradient dispersion.
>
> - **Variance:** The residual layer has minimal impact on training variance for PPO-res and DPO-res. This is because their gradient calculation mechanisms are similar to the original PPO and DPO, limiting the residual layer's variance reduction effect.
>
> In contrast, for policies with inherently high variance, such as KPN and AdvKPN, the integration of a residual layer significantly stabilizes training, as highlighted in our theoretical analysis and supported by visualizations in Appendix C. This underscores the residual layer's effectiveness in addressing variance-related instability in RKHS-based methods.
>
> We hope these additional experiments and explanations address your concerns about the fairness of the comparisons and provide further clarity on the impact of our residual design. Regarding the **discussion of the variance reduction methods introduced in previous works**, We have added a comprehensive discussion in Appendix D.1, comparing existing variance reduction methods and highlighting the computational challenges these approaches face when applied to the problems we address. Regarding the variance reduction in our final algorithm, ResKPN, we would like to emphasize two key points:
>
> (a) ResKPN integrates many widely used variance reduction techniques in reinforcement learning. For example, it incorporates the Advantage function (discussed in Section 3.3) and utilizes general variance reduction techniques commonly employed in PPO and DPO algorithms (explained in Appendix B.1). Therefore, our approach builds upon and combines several existing variance reduction methods.
>
> (b) The residual design in ResKPN offers a significant advantage by effectively reducing the variance of RKHS policies through the addition of a residual layer in the representation learning component. This reduction is achieved without requiring complex algorithmic implementation. We argue that the residual design is not mutually exclusive with other variance reduction techniques; rather, it can complement these methods to potentially achieve even better results.
>
> We hope this explanation clarifies how ResKPN incorporates and complements existing variance reduction techniques.

---

> ### Author Response · Authors · 2024-11-18
> **Response to Reviewer eyTM (5/5)**
>
> > **2. There is existing literature on combining RKHS with residual networks, and a discussion of these studies would add valuable context.**
>
> response: Thank you for your valuable suggestion. We have reviewed the existing literature on combining RKHS with residual networks and acknowledge its relevance. While such studies exist, they have not been applied to the reinforcement learning domain. To provide context, we have added a discussion in Section 2.2, briefly highlighting the role of RKHS in explaining the smoother functions learned by residual networks and their superior generalization capabilities. We hope this addition addresses your concern and provides valuable context to our work.
>
> > **3. The numerical section would benefit from testing in more complex environments to strengthen the evaluation.**
>
> response: Thank you for your insightful feedback. To enhance our evaluation, we have included two additional environments: **Pusher** and **Reacher**. The **Pusher** environment involves controlling a robotic arm to push an object to a target, requiring precise coordination and object manipulation. The **Reacher** environment challenges the agent to move a robotic arm’s end-effector to a target, emphasizing precision in control and sensitivity to rewards. These environments provide a broader evaluation of our proposed algorithms. The added experiments are presented in Appendix D.2.
>
> In the Pusher environment, ResKPN consistently achieves the best performance, while in the Reacher environment, all algorithms except KPN and Origin-Kernel converge to the optimal reward. Additionally, ResKPN demonstrates superior stability, with minimal variance across training episodes in both environments. These results further validate ResKPN’s robustness and adaptability to diverse and precise control tasks.
>
> Regarding the complexity of previously tested environments, we note that our evaluation already includes highly challenging benchmarks such as **Humanoid** and **HumanoidStandup**, with state spaces of dimensionality $ \mathbb{R}^{348} $ and action spaces of $ [-0.4, 0.4]^{17} $. These environments are among the most demanding in single-agent reinforcement learning, requiring significant computational and algorithmic capability.
>
> We acknowledge that scalability to multi-agent settings remains a potential challenge, and extending our framework to address these scenarios is a key focus of our future work. Thank you again for your valuable suggestions, which have helped us strengthen the evaluation and discussion of our manuscript.
>
> > **References for our response (including references from reviewer):**
> >
> [1] Mazoure, B., Doan, T., Li, T., Makarenkov, V., Pineau, J., Precup, D., & Rabusseau, G. (2020). Representation of reinforcement learning policies in reproducing kernel hilbert spaces. arXiv preprint arXiv:2002.02863.
>
> [2] Wang, Y., & Principe, J. C. (2021). Reinforcement learning in reproducing kernel Hilbert spaces. IEEE Signal Processing Magazine.
>
> ---
> [3] Paternain, S., Bazerque, J. A., Small, A., & Ribeiro, A. (2020). Stochastic policy gradient ascent in reproducing kernel hilbert spaces. IEEE Transactions on Automatic Control.
>
> [4] Le, T. P., Ngo, V. A., Jaramillo, P. M., & Chung, T. (2019). Importance sampling policy gradient algorithms in reproducing kernel hilbert space. Artificial Intelligence Review.
>
> [5] Qian, X., & Klabjan, D. (2020). The impact of the mini-batch size on the variance of gradients in stochastic gradient descent. arXiv preprint arXiv:2004.13146.
>
> [6] Liu, J., & Lian, H. (2024). Kernel-Based Decentralized Policy Evaluation for Reinforcement Learning. IEEE Transactions on Neural Networks and Learning Systems.

---

### Author Response · Authors · 2024-11-19
**Global response**

We would like to express our sincere gratitude to all the reviewers for their thorough and insightful feedback. Your valuable comments and suggestions have significantly contributed to enhancing the quality and clarity of our paper. Based on your inputs, we have made several improvements to our manuscript to address the concerns raised and to better highlight the contributions of our work. Below is a summary of the modifications we have implemented:

1. **Expanded the Discussion of Related Work and Literature Review:**
   - Added discussions of additional references in the **Introduction (Section 1)** and **Section 2.2**, and provided detailed comparisons with our work.
   - Expanded the discussion on the application of existing RKHS in reinforcement learning, especially regarding variance reduction techniques.
   - Added discussions in **Section 2.2** on existing studies that combine RKHS with residual networks, providing more comprehensive background information and highlighting the innovation of our work in this field.
   - Added **Table 2** in **Appendix D.1**, summarizing variance reduction methods for RKHS policies, to provide a clearer comparison.

2. **Simplified and Enhanced the Variance Analysis Section:**
   - Introduced new notations (e.g., $\Gamma_A$, $\Gamma_V$, $\Gamma_{h,\mu}$) in the variance analysis section to simplify mathematical formulas and improve readability.
   - Added more intuitive explanations and insights after **Theorem 3.2** to help readers better understand how the residual neural network reduces the variance of RKHS policies.
   - Included visual explanations and illustrations in **Appendix C** to enhance the intuitiveness of the theoretical analysis.

3. **Added Experimental Results and Additional Experiments:**
   - **Fairness Comparison Experiments**: Added experiments in **Appendix D.3** examining the impact of the residual layer on PPO and DPO algorithms, exploring the fairness and effectiveness of the residual design.
   - **Complex Environment Testing**: Included experiments in **Appendix D.2** on two more complex environments, **Pusher** and **Reacher**, to verify the applicability and robustness of our algorithm.
   - **Minibatch Gradient Experiments**: Conducted experiments on minibatch gradients in **Appendix D.1**, exploring their impact on the variance of RKHS policies and analyzing computational costs.
   - **Experiments with Different Kernels**: Added experiments using Laplacian, Sigmoid, and Linear kernels in **Appendix D.4** and **Appendix E**, comparing the trade-off between performance and computational cost.
   - **Computational Cost Analysis:** Added a table in **Appendix E** detailing the training times under different kernels and environments, providing a quantitative analysis of the algorithm's computational cost.

4. **Corrected and Clarified Descriptions in the Paper:**
   - Corrected the Description of Function \( h \): Clarified that \( h \) is a function, not a functional, correcting the erroneous description on line 106.
   - Updated the Description of Figure 1a, correcting the previous error.
   - Updated Figure 1: Tested both subplots (a) and (b) in the same environment, enhancing the persuasiveness and consistency of the figure, and updated the corresponding description.
   - Moved and Updated Figure 6 (now Figure 2): Moved it into the main text to provide a detailed illustration of the algorithm's structure, and integrated the Critic network into the figure to offer a more comprehensive overview.
5. **Added Discussions on Limitations and Future Work:**
   - Included discussions in the **Conclusion section** on the limitations of RKHS policies in computational cost and scalability, and outlined future work directions such as kernel embedding techniques and efficient kernel approximations.
   - Discussed the potential extension of the algorithm to **multi-agent and cooperative environments**, analyzed the challenges and applicability, and elaborated on related computational cost and scalability issues in **Appendix E**.

We trust that these revisions adequately address the reviewers' concerns and enhance the overall quality of our paper. Additionally, we have uploaded our project code as supplementary material to facilitate the reproducibility of our work.

Thank you once again for your valuable feedback!

---

### Meta-Review · Area_Chair_9fvW · 2024-12-17

**Metareview:**

This paper aims to improve the stability and robustness of RKHS-based policy gradient methods. A residual connection is introduced to achieve variance reduction. The paper is theoretically grounded and supported by extensive experiments.

The reviewers commented that the paper is technically well founded, the experiments are convincing and comprehensive, and the presentation is clear.

Some reviewers expressed concerns such as lack of discussions with respect to previous works on RKHS-based RL methods and variance reduction, but the authors have added sufficient corresponding discussions. The authors have also addressed some other questions raised by reviewers, such as simplifying the variance analysis via extra notations, adding better illustration of the high variance problem, and showing the computational time needed by the algorithm. Another concern expressed by a reviewer is the lack of open-sourced code, but this has also been addressed by authors since the code was later uploaded to the supplemental material.

Overall, the reviewers think that the paper makes important contributions to the field of RKHS-based RL both theoretically and empirically. So, acceptance is recommended.

**Additional Comments On Reviewer Discussion:**

The authors provided extensive responses to the questions from each of the reviewers, and managed to address their concerns.

---

### Decision · Program_Chairs · 2025-01-22

Accept (Poster)